# A remarkable new deep-sea nereidid (Annelida: Nereididae) with gills

Tulio F. Villalobos-Guerrero[1], Sonja Huč[2], Ekin Tilic[2,3], Avery S. Hiley[2], Greg W. Rouse[2]*

**1** Department of Marine Ecology, Centro de Investigación Científica y de Educación Superior de Ensenada, Ensenada, Baja California, Mexico, **2** Scripps Institution of Oceanography, University of California San Diego, La Jolla, La Jolla, California, United States of America, **3** Department of Marine Zoology, Senckenberg Research Institute and Natural History Museum, Frankfurt am Main, Germany

* grouse@ucsd.edu

**Data Availability Statement:** All relevant data are within the manuscript and its Supporting Information files. Sequences are available via GenBank. The accession codes are within the paper.

## Abstract

Nereidid polychaetes are well known from shallow marine habitats, but their diversity in the deep sea is poorly known. Here we describe an unusual new nereidid species found at methane seeps off the Pacific coast of Costa Rica. Specimens of *Pectinereis strickrotti* gen. nov., sp. nov. had been observed dating back to 2009 swimming just above the seafloor at ~1,000 m depth but were not successfully captured until 2018. Male epitokes were collected as well as a fragment of an infaunal female found in a pushcore sample. The specimens were all confirmed as the same species based on mitochondrial COI. Phylogenetic analyses, including one based on available whole mitochondrial genomes for nereidids, revealed no close relative, allowing for the placement of the new species in its own genus within the subfamily Nereidinae. This was supported by the unusual non-reproductive and epitokous morphology, including parapodial cirrostyles as pectinate gills, hooked aciculae, elfin-shoe-shaped ventral cirrophores, and elongate, fusiform dorsal ligules emerging sub-medially to enlarged cirrophores. Additionally, the gill-bearing subfamily Dendronereidinae, generally regarded as a junior synonym of Gymnonereidinae, is reviewed and it is here reinstated and as a monogeneric taxon.

## Introduction

The diversity of Nereididae de Blainville, 1818 compared to other polychaete clades of family rank is well documented, with over 700 accepted species [1–4]. Nereidids are generally known from coastal regions, commonly confined to shallow marine habitats, although they also occur in brackish, freshwater, and even moist terrestrial environments [1, 5–8]. However, ~10% of the total diversity is known from deep-sea habitats. Currently, 69 nereidid species found in 13 genera are known from below 500 m depth [9] from various habitats, including hydrothermal vents, cold seeps, polymetallic nodules, foraminifera ooze, whale carcasses, and sunken wood [10–15]. The best-represented genera are the polyphyletic *Neanthes* Kinberg, 1865 (16 species) and *Nereis* Linnaeus, 1758 (19 species), which includes the deepest known nereidid recorded,

**Funding:** GWR NSF-OCE 0939557 US National Science Foundation https://www.nsf.gov. The funders had no role in study design, data collection and analysis, decision to publish, or preparation of the manuscript.

**Competing interests:** The authors have declared that no competing interests exist.

*Nereis profundi* Kirkegaard, 1956 from 7,290 m depth. One genus, the monotypic *Typhlonereis* Hansen, 1879 is exclusive to deep environments.

The deep sea encompasses a diverse, vast mosaic of understudied and poorly sampled habitats [16]. Several surveys that have sampled deep waters from distinct regions suggest polychaete biodiversity is severely underestimated [17–19], rendering our knowledge of it limited [20–22]. We can assume, therefore, that oceanic depths still host a vast number of yet undescribed nereidid species, although it is noteworthy that there has been a relatively continual discovery of new deep-sea nereidids since about the 1960s (e.g., [6, 10–12, 14, 15, 23–36].

While deep-water nereidids continue to be discovered, information about their behavior and inference about morphological adaptations has been scarce. According to Fauchald [11, 12], deep-sea nereidids share a few unusual prostomial and parapodial features when compared to shallow-living relatives, such as the absence of eyes, prolonged appendages and chaetae, and posterior chaetigers with extended notopodia and elongated neuropodia. Notably, the reduction of eyes and the elongation of parapodial appendages and chaetae fall under the general specialization to aphotic environments of some subterranean and cave (troglobiotic) nereidids placed in Namanereidinae Hartman, 1959 [37–39]. These morphological adaptations, inherent to the 'darkness syndrome' [40], have been shown for a few other polychaete clades containing cave and deep-sea members (e.g., Scalibregmatidae Malmgren, 1867 [41] and Aphroditiformia Levinsen, 1883 [42–45]), suggesting that these are also convergent evolutionary traits for nereidids living at aphotic depths.

Genetic tools have been instrumental in developing a better understanding of the diversity among nereidids. Several new species or previously synonymized ones have been described, reinstated, or delimited through the integration of morphological data and molecular markers of apparently cryptic species (*e.g.*, [46–57]). Molecular data has also been utilized to investigate phylogenetic relationships at a broader scale among nereidid taxa [15, 57, 58]. Recently mitochondrial genomes (mitogenomes) have been sequenced for a variety of nereidid species (e.g., [59–66], providing a rich dataset to assess phylogenetic relationships with more confidence. Alves et al. [67] assessed the monophyly and phylogenetic relationships of the presently accepted subfamilies and provided an ancestral state reconstruction of pharyngeal structures using mitogenomic data. They rejected the monophyly of the subfamilies Nereidinae de Blainville, 1818 and Gymnonereidinae Banse, 1977 as currently recognized and revealed that the occurrence of papillae and paragnaths may not be reliable features to diagnose major groups. We revisit the nereidid subfamily issue here with regards to Dendronereidinae Pillai, 1961.

This study describes a new, morphologically unusual, nereidid species belonging to a new genus using specimens found near methane seeps at ~1,000 m depth off Costa Rica in the eastern Pacific. This new species is remarkable for its dorsal and ventral anterior parapodial cirri, modified as gills, and hooked-shaped posterior aciculae, both unique features among Nereididae. Its mitochondrial genome was sequenced as well as those of two shallow-water nereidid species. This allowed for a new mitogenomic analysis to evaluate the position and relationships of the new genus within Nereididae.

## Material and methods

### Specimen collection and preparation

Nereidid specimens were repeatedly seen swimming just above the seafloor at ~1,000 m depth off Costa Rica (Eastern Tropical Pacific) on cruises dating back to 2009 (Fig 1 and S1 Video). However, it was not until 2018 that three epitokous males and one fragmented infaunal female were successfully collected near methane seeps of Mound 12 via the submersible DSV (deep submergence vehicle) *Alvin*, operated from the RV *Atlantis* (Fig 1B–1D and S1 Video). The

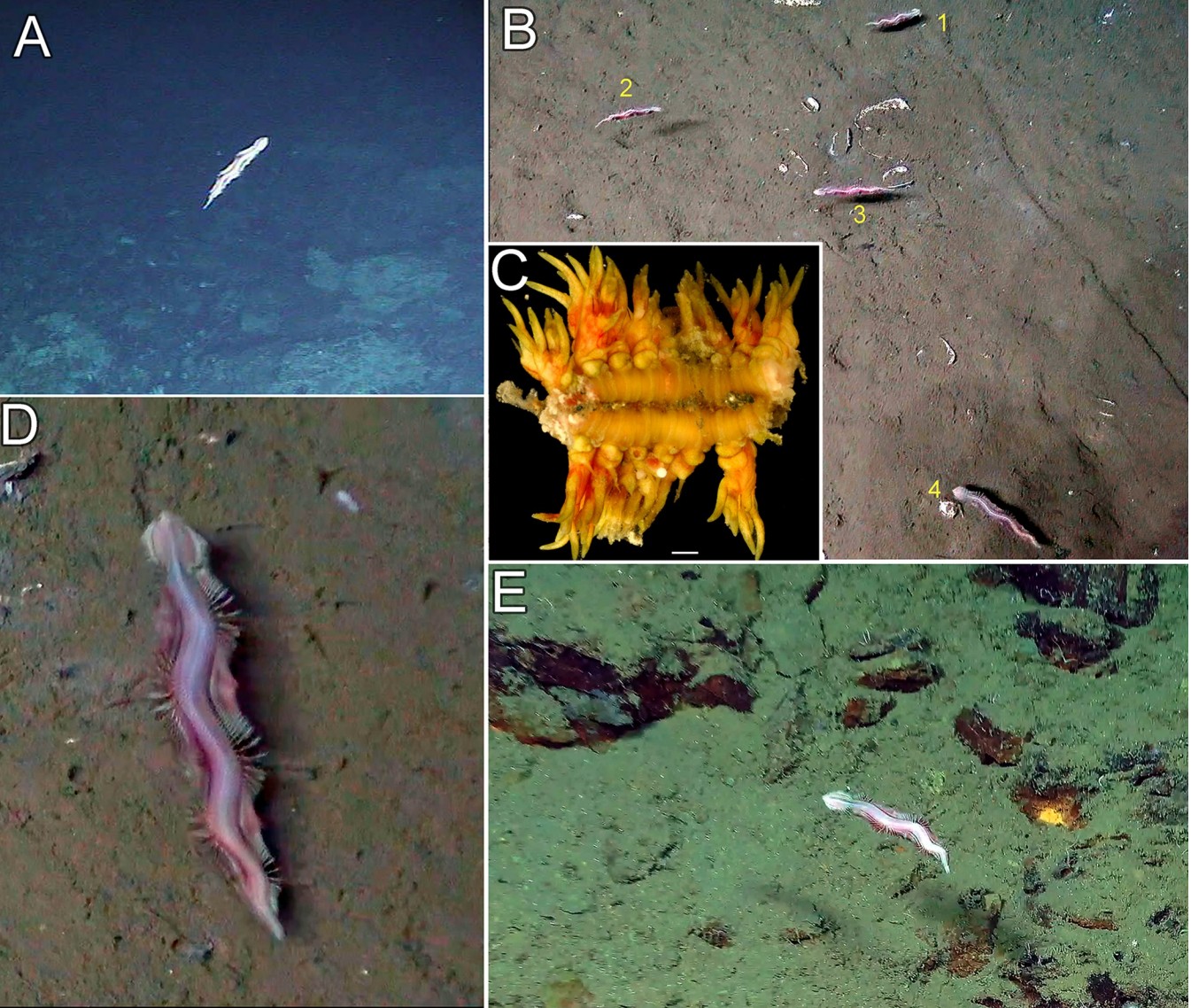

**Fig 1. *Pectinereis strickrotti* gen. nov., sp. nov. in life.** A, B, D. Several epitokous males swimming near methane seeps of Mound 12 (~1,000 m depth) of the Costa Rica margin and videoed via the submersible DSV *Alvin*. A. A frame grab from a video taken on *Alvin* dive 4503 on Feb. 4, 2009. B and D. Frame grabs from video taken on *Alvin* dive 4987 on Nov. 2, 2018. C. A fragment of an atokous infaunal female was collected at the same depth and locality via sediment pushcore on *Alvin* dive 4984 on Oct. 30, 2018. A white egg ~350 μm in diameter is visible on the exterior. Scalebar 1 mm. E. An epitokous male swimming near methane seeps of Parrita Scar (~1,000 m depth) of the Costa Rica margin. The specimen was initially caught via slurp with the ROV *SuBastian* (dive S0218, Jan. 11, 2019) but escaped. Images A, B, D, courtesy of Woods Hole Oceanographic Institute. E, courtesy of Schmidt Ocean Institute.

epitokes were suction sampled while swimming just above the bottom. The female fragment was incidentally cored with a PVC sediment push core. One specimen was also observed on a cruise in 2019 to the same region by the remotely operated vehicle (ROV) *SuBastian* operated from the RV *Falkor* (Fig 1E and S2 Video). See Levin *et al*. [13, 68] for details about location sites. The type specimens are deposited at the Benthic Invertebrate Collection, Scripps Institution of Oceanography, La Jolla, California, USA (SIO-BIC) and the Museo de Zoología (Universidad de Costa Rica), San José, Costa Rica (MZ-UCR). Specimens were collected under the following permits issued by CONAGEBIO (Comisión Nacional para la Gestión de la Biodiversidad) and SINAC (Sistema Nacional de Áreas de Conservación) under MINAE (Ministerio

de Ambiente y Energía), Government of Costa Rica: SINAC- CUSBSE-PI-R-032-2018 and the Contract for the Grant of Prior Informed Consent between MINAE-SINAC-ACMC and Jorge Cortés Nuñez for the Basic Research Project "Cuantificación de los vínculos biológicos, quími- cos y físicos entre las comunidades quimiosintéticas con el mar profundo circundante.'

## DNA extraction, sequencing, and genome skimming

DNA from each of the available specimens of the new species, as well as representatives of *Nec- toneanthes oxypoda* (von Marenzeller, 1879) (SIO-BIC A13109) and *Nereis pelagica* Linnaeus, 1758 (SIO-BIC A6054) was extracted using the Zymo Research Quick-DNA™ Miniprep Plus kit, following the protocol supplied by the manufacturer. Mitochondrial cytochrome c oxidase subunit I (*COI*) sequences were obtained from the specimens via Sanger sequencing, with PCR amplification carried out using a mixture of 12.5 µl Apex 2.0x Taq Red DNA Polymerase Mas- ter Mix (Genesee Scientific, San Diego, California, USA), 1 µl each of the primers LCO1490/ HCO2198 [69] (10 µM), 8.5 µl of ddH2O, and 2 µl of eluted DNA. A Mastercycler 5345 epGra- dient thermal cycler (Eppendorf, Hamburg, Germany) was used with the reaction protocol 94˚C/180s –(94˚C/30s – 47˚C/45s – 72˚C/60s) * 5 cycles–(94˚C/30s – 52˚C/45s – 72˚C/60s) * 30 cycles– 72˚C/300s. Final PCR products were purified with the ExoSAP-IT protocol (USB Affymetrix, Ohio, USA), and sequencing was performed by Eurofins Genomics (Louisville, Kentucky, USA). Sequences were assembled using Geneious v. 2022.2.2 (©Biomatters Ltd.; http://www.geneious.com/, New Zealand), and the new sequences were deposited to GenBank (OQ415952-OQ415955).

Genome skimming was conducted on the holotype of the new species, *Nectoneanthes oxy- poda*, and *Nereis pelagica*. Prior to genome skimming, total DNA concentration for gDNA extractions was estimated using the Qubit dsDNA BR Assay Kit with a Qubit fluorometer (Invitrogen), and DNA quality was assessed with agarose gel electrophoresis at 100 V for 65 min. The DNA extractions were sent to Novogene (en.novogene.com) for library preparation and whole genome sequencing using 150 base pair (bp) paired-end reads on the Illumina NovaSeq 6000 platform (Illumina, San Diego, CA), resulting in 9.8–13.3 million paired-end raw reads per sample.

## Mitochondrial genome assembly and annotation

Sequence reads were trimmed and cleaned with Trimmomatic v. 0.39 [70] before assembly with MitoFinder v. 1.4 [71] with The Invertebrate Mitochondrial Code (NCBI; transl_table = 5) used to translate the 13 protein-coding genes (PCGs). Complete records for all RefSeq annelid mitogenomes available on NCBI GenBank were used as the MitoFinder reference file, with MEGAHIT v. 1.2.9 [72] metagenomic assembler and Arwen v.1.2.3 [73] tRNA gene annotator selected for the assembly parameters. Resulting mitochondrial genes recovered in MitoFinder [71] contigs were checked for contamination using NCBI's Nucleotide BLAST. The MITOS Web Server [74] was used for mitogenome annotation. Geneious Prime v. 2022.2.2 [75] was used to manually finalize annotations, extract the 13 PCGs and two ribosomal RNA genes (rRNAs), and translate the PCGs into amino acids. Nuclear *18S* rRNA gene sequences (*18S*) were also mined out for the three species: using BBMap v. 38.87 [76], post-Trimmomatic paired-end reads were interleaved with the included reformat.sh script. Publicly available and closely related nereidid *18S* sequences were extracted in FASTA format from NCBI. Inter- leaved FASTQ files for each of the three species were mapped individually to the nereidid 18S sequences using Minimap2 v. 2.22 [77, 78], then SAMtools v. 1.13 [79] was used to extract the resulting mapped reads. The Map to Reference tool in Geneious Prime v. 2022.2.2 [75] was used to individually map the FASTQ mapped reads to the nereidid *18S* FASTA file for read

coverage visualization and consensus sequence extraction. Newly assembled and annotated mitogenomes obtained in this study were deposited in GenBank with accession numbers OL782598-600 and the 18S sequences as OR437939-41 (Table 1 and S1 Table).

## Haplotype network and phylogenetic analyses

A haplotype network using *COI* data from the four specimens of the new species was generated with PopART v. 1.7 [80] using the TCS algorithm [81, 82]. A maximum likelihood (ML) phylogenetic analysis of concatenated *COI*, *16S rRNA*, and *18S rRNA* DNA sequences from across Nereididae (S1 Table) was analyzed, with each gene under the model GTR+FO+I+G4, chosen with ModelTest-NG v. 0.1.7 [83] and executed using raxmlGUI v. 2.0.10 [84]. The best ML tree was chosen after 100 ML runs (seed 581027) and support was assessed via thorough bootstrapping (1,000 pseudoreplicates).

For the mitogenome-based phylogeny, alignments of the three newly generated mitochondrial genomes (two *rRNAs* and 13 PCGs translated into amino acids) along with relevant data on GenBank from several different studies (Table 1) were performed in Mesquite v. 3.61 [85] for each gene using the MUSCLE [86] algorithm with default settings. Data were partitioned by gene, with best-fit models for these partitions selected using ModelTest-NG v. 0.1.7. The substitution models selected were MTZOA for *ATP6*, *COI*, *COII*, *COIII*, *CYTB*, *ND1*, *ND3*, *ND4*, and *ND5*; MTMAM for *ND2*, *ND4L*, and *ND6*; MTREV for *ATP8*; TIM2 +I +G for *12S*; and TIM2 +G for *16S*. The data was then analyzed via an ML analysis with search and bootstrap parameters as with the three-gene analysis. Following Alves et al. [67], the terminals from Chrysopetalidae Ehlers, 1864 (*Arichlidon* Watson Russell, 1998 and *Bhawania* Schmarda, 1861), Microphthalmidae Hartmann-Schröder, 1971 (*Microphthalmus* Mecznikow, 1865 and *Hesionides* Friedrich, 1937), and Hesionidae Grube, 1850 (*Oxydromus* Grube, 1855) were used as outgroups.

## Morphology

Methods for measurements of specimens (total body length, TL; length to chaetiger 15, L15; body width to chaetiger 15, W15), counting of structures and ridge patterns on the proboscis, and body dissections were explained elsewhere [87]. For practical purposes, decimal numbers are used when measurements between two structures exceed one unit (*e.g.*, 1.3 times, 2.5 times, twice). In contrast, written fractions were used when those measurements were less than one unit (e.g., half, two-thirds, four-fifths).

Light microscopy observations were made using both stereo and compound microscopes. Specimens were photographed alive using a Canon EOS M5 camera. Preserved specimens were photographed with a Nikon D5100 or Canon Rebel T7 camera mounted on both the compound microscope and stereomicroscope. Some images were generated using stacks using Helicon Focus® 6 (Method C) or manually through Adobe Photoshop® CS6 (for chaetae). Parapodia of one specimen were processed for scanning electron microscopy (SEM) with a Zeiss Evo10 scanning electron microscope. Figure backgrounds were cleaned and darkened or lightened as necessary without manipulating the actual specimen. Parapodia were shown in anterior views unless otherwise stated.

Descriptions of the species are based on the holotype morphology unless otherwise stated. The methods performed, terminology, and standardized definitions established for overall nereidid features either newly proposed, partially readapted, or based upon references cited in Villalobos-Guerrero et al. [4] were followed. These authors proposed a division of the dorsal ligule into two main regions: proximal and distal. However, we consider that the proximal dorsal ligule ([4]: Fig 1C, dld) is instead the basal part of dorsal cirrus, namely the dorsal

**Table 1. Mitochondrial sequences used for the phylogenetic analysis.** Three new mitogenomes were generated for this study (**bold**). Also, four COI sequences generated for the holotype and three paratypes of *Pectinereis strickrotti* gen. nov., sp. nov. are listed. Note that the name *Laeonereis* cf. *pandoensis* is used here instead of *Laeonereis culveri* (Webster, 1879) since the specimen was collected in Brazil. The correct spelling for sequences deposited in GenBank as *Tylorrhynchus heterochaetus* is actually *Tylorrhynchus heterochaetus*.

| Taxon | Citation/Voucher | Collection Site | GenBank Accession Numbers |
|---|---|---|---|
| *Alitta succinea* (Leuckart, 1847) | Alves et al. (2020) | USA: Florida | MN812981 |
| *Arichlidon gathofi* Watson Russell, 2000 1 | Alves et al. (2020) | USA: North Carolina | MN855126 (COI), MN855135 (COX2), MN855144 (COX3), MN855116 (CYTB), MN855107 (ATP6), MN855196 (ND5), MN855188 (ND4L), MN855178 (ND4), MN855154 (ND1), MN855171 (ND3), MN855164 (ND2) |
| *Arichlidon gathofi* Watson Russell, 2000 2 | Alves et al. (2020) | Panama: Bocas del Toro | MN855127 (COI), MN855136 (COX2), MN855145 (COX3), MN855205 (ND6), MN855117 (CYTB), MN855108 (ATP6), MN855197 (ND5), MN855179 (ND4), MN855155 (ND1), MN855172 (ND3), MN855165 (ND2) |
| *Bhawania goodei* Webster, 1884 | Alves et al. (2020) | Panama: Bocas del Toro | MN855128 (COI), MN855146 (COX3), MN855118 (CYTB), MN855198 (ND5), MN855189 (ND4L), MN855180 (ND4), MN855208 (16S), MN855156 (ND1), MN855173 (ND3), MN855166 (ND2) |
| *Cheilonereis cyclurus* (Harrington, 1897) | Park et al. (2017) | South Korea: Gangwondo | MF538532 |
| *Dendronereis chipolini* Hsueh, 2019 | Zhen et al. (2022) | China: Beibu Gulf | MW532084 |
| *Hediste diadroma* Sato & Nakashima, 2003 | Kim et al. (2016) | South Korea: Masan Port | KX499500 |
| *Hediste japonica* (Izuka, 1908) | Park et al. (2020) | South Korea: Incheon | MN876864 |
| *Hesionides* sp. | Alves et al. (2020) | Panama: Bocas del Toro | MN855129 (COI), MN855137 (COX2), MN855147 (COX3), MN855119 (CYTB), MN855109 (ATP6), MN855199 (ND5), MN855190 (ND4L), MN855181 (ND4), MN855157 (ND1), MN855174 (ND3), MN855167 (ND2) |
| *Laeonereis* cf. *pandoensis* | Seixas et al. (2016) | Brazil: Rio de Janeiro | KU992689 |
| *Microphthalmus listensis* Westheide, 1967 1 | Alves et al. (2020) | Germany: Sylt | MN855130 (COI), MN855138 (COX2), MN855148 (COX3), MN855206 (ND6), MN855120 (CYTB), MN855110 (ATP6), MN855182 (ND4), MN855209 (16S), MN855158 (ND1) |
| *Microphthalmus listensis* Westheide, 1967 2 | Alves et al. (2020) | Germany: Sylt | MN855139 (COX2), MN855149 (COX3), MN855121 (CYTB), MN855111 (ATP6), MN855200 (ND5), MN855191 (ND4L), MN855183 (ND4), MN855210 (16S), MN855159 (ND1), MN855175 (ND3) |
| *Microphthalmus similis* Bobretzky, 1870 | Alves et al. (2020) | Germany: Sylt | MN855131 (COI), MN855140 (COX2), MN855150 (COX3), MN855122 (CYTB), MN855112 (ATP6), MN855201 (ND5), MN855192 (ND4L), MN855184 (ND4), MN855160 (ND1), MN855168 (ND2) |
| *Namalycastis abiuma* (Grube, 1872) | Lin et al. (2016) | China: Xiamen | KU351089 |
| *Neanthes glandicincta* (Southern, 1921) | Lin et al. (2017) | China: Xiamen | KY094478 |
| **Nectoneanthes oxypoda (Marenzeller, 1879)** | **SIO-BIC A13109** | **Japan: Osaka Bay** | **OL782599** |
| **Nereis pelagica Linnaeus, 1758** | **SIO-BIC A6054** | **Norway: Trondheim** | **OL782598** |
| *Nereis* sp. | Kim et al. (2017) | South Korea: Dok-do Isl. | MF960765 |
| *Nereis zonata* Malmgren, 1867 | Nam et al. (2021) | Beaufort Sea | MT980928 |
| *Oxydromus pugettensis* (Johnson, 1901) | Alves et al. (2020) | USA: Washington | MN855132 (COI), MN855141 (COX2), MN855151 (COX3), MN855123 (CYTB), MN855113 (ATP6), MN855202 (ND5), MN855193 (ND4L), MN855184 (ND4), MN855211 (16S), MN855161 (ND1), MN855176 (ND3) |
| *Oxydromus* sp. | Alves et al. (2020) | Panama: Bocas del Toro | MN855133 (COI), MN855142 (COX2), MN855152 (COX3), MN855124 (CYTB), MN855114 (ATP6), MN855203 (ND5), MN855194 (ND4L), MN855186 (ND4), MN855212 (16S), MN855162 (ND1), MN855177 (ND3), MN855169 (ND2) |
| *Paraleonnates uschakovi* Chlebovitsch & Wu, 1962 | Park et al. (2016) | South Korea: Ganghwa Isl. | KX462988 |
| **Pectinereis strickrotti gen. nov., sp. nov.** | **SIO-BIC A9836 (Holotype)** | **Costa Rica (Pacific): Mound 12** | **OL782600, (OQ415952-5 COI of holotype and the 3 paratypes)** |
| *Perinereis aibuhitensis* (Grube, 1878) | Kim et al. (2015) | South Korea: Ganghwa Isl. | KF611806 |

*(Continued)*

**Table 1.** (Continued)

| Taxon | Citation/Voucher | Collection Site | GenBank Accession Numbers |
|---|---|---|---|
| *Perinereis cultrifera* (Grube, 1840) | Alves et al. (2020) | France: Arcachon | MN812983 |
| *Perinereis nuntia* (Lamarck, 1818) | Won et al. (2013) | South Korea: Yeosu | JX644015 |
| *Perinereis* sp. | Alves et al. (2020) | Panama: Bocas del Toro | MN823962 (COI), MN823963 (COX2), MN823964 (COX3), MN823970 (ND6), MN823961 (CYTB), MN823960 (ATP6), MN823969 (ND5), MN823968 (ND4L), MN823967 (ND4), MN823972 (12S), MN823971 (16S), MN823965 (ND1), MN823966 (ND2) |
| *Platynereis bicanaliculata* (Baird, 1863) | Alves et al. (2020) | USA: Washington | MN812984 |
| *Platynereis* cf. *australis* | Alves et al. (2020) | Chile: Chonchi | MN830367 |
| *Platynereis dumerilii* (Audouin & Milne Edwards, 1833) | Boore & Brown (2000) | Europe | AF178678 |
| *Platynereis massiliensis* (Moquin-Tandon, 1869) | Alves et al. (2020) | Wales: West Angle Bay | MN812985 |
| *Platynereis* sp. 1 | Alves et al. (2020) | Brazil: Ceara | MN830365 |
| *Platynereis* sp. 2 | Alves et al. (2020) | Brazil: Rio de Janeiro | MN830366 |
| *Pseudonereis variegata* (Grube, 1857) | Alves et al. (2020) | South Africa: Western Cape | MN855134 (COI), MN855143 (COX2), MN855153 (COX3), MN855207 (ND6), MN855125 (CYTB), MN855115 (ATP6), MN855204 (ND5), MN855195 (ND4L), MN855187 (ND4), MN855214 (12S), MN855213 (16S), MN855163 (ND1), MN855170 (ND2) |
| *Tylorrhynchus heterochaetus* (sic) (Quatrefages, 1866) | Chen et al. (2016) | China: Nalong River | KM111507 |

cirrophore, and the dorsal cirrus ([4]: Fig 1C, dc) is, therefore, the dorsal cirrostyle; whereas the distal region ([4]: Fig 1C, dlp) corresponds to the dorsal ligule itself. This is grounded on the circulatory system's arrangement in some nereidids' enlarged notopodia. The placement and form of the notopodial vessels running alongside the lateral margins of such an enlarged structure are here assumed to be homologs. These have distinctly been illustrated in *A. succinea* Leuckart, 1847 (see [88]: 75, Fig 4G and 4J, as *Nereis limbata*), *Namalycastis abiuma* (Grube, 1872) (see [89]: 23, Fig 6, arrows in dorsal cirri) and *Stenoninereis* species (see [90]: 98, Fig 1B, solid red lines). This presumed homology is also evident in several other Nereidinae, Gymnonereidinae, and Namanereidinae members, such as *A. acutifolia* (Ehlers, 1901) ([49]: 168, 3E, F), *Dendronereis aestuarina* Southern, 1921 ([91]: Pl. 20, Fig 4E and 4F), *Gymnonereis sibogae* (Horst, 1918) ([92]: 39, Fig 32b), *Namalycastis borealis* Glasby, 1999 ([38]: 33, Fig 5H), *Namanereis occulta* (Conde-Vela, 2013) ([38]: 33, Fig 5G), *Neanthes micromma* (Harper, 1979) ([93]: 100, Fig 6), and among others. All these taxa share the distal placement of the dorsal cirrostyle, which is easily recognized by the articulation and thickened basal tegument (sometimes barely evident in *Namalycastis*), and the reduced dorsal ligule when present on the fully expanded dorsal cirrophores. Nicoll [88] and Kaufmant [94] also found a similar irrigation of the notopodium in *A. virens* Sars, 1834 and *Hediste diversicolor* (Müller, 1776) (both as *Nereis*), respectively. However, the position of notopodial vessels and the innervations of capillaries were different; the dorsal cirrophore was only slightly enlarged, and the dorsal cirrostyle was sited sub-medially on the notopodia.

Finally, the relative extension of parapodial structures was described following Villalobos-Guerrero & Carrera-Parra [49]. However, the dorsal ligule and the parapodial cirrostyles were measured in comparison with the entire length of the parapodial cirrophore in natatory chaetigers of epitokous specimens. The first and last natatory chaetigers of epitokes were determined by the appearance/disappearance of additional parapodial lobes, particularly the expanded neuropodial postchaetal lobe.

## Nomenclatural acts

The electronic edition of this article conforms to the requirements of the amended International Code of Zoological Nomenclature, and hence the new names contained herein are available under that Code from the electronic edition of this article. This published work and the nomenclatural acts it contains have been registered in ZooBank, the online registration system for the ICZN. The ZooBank LSIDs (Life Science Identifiers) can be resolved, and the associated information is viewed through any standard web browser by appending the LSID to the prefix "http://zoobank.org/". The LSID for this publication is: urn:lsid:zoobank.org:pub:9E9C5C-D6-EFE8-4B90-A63F-96A4544AC60B.

## Results

### Haplotype network and phylogenetic analysis

The four *COI* sequences obtained from the new species were all unique, but varied by a maximum of less than 1%, five base pairs out of an alignment of 676 bases (Fig 2A). The fragment of an atokous infaunal female (Fig 1C) was clearly the same species as the epitokous males (Fig 1A, 1B and 1D) and differed by only 3–5 base pairs. The ML phylogenetic analysis (log likelihood = -39786.879947) based on the concatenated *COI*, *16S*, and *18S* DNA dataset of 4,320 bases (Fig 3) showed the new taxon under study here (*Pectinereis strickrotti* gen. nov., sp. nov.) with no well-supported close relationships among the other Nereididae but was well nested within a strongly supported Nereidinae (Fig 3). It did form a clade with a *Hediste* terminals but with low support. *Paraleonnates* was the sister group to Nereidinae, though with moderate support, with Gymnonereidinae as sister group to this clade, though with low support. *Tylorrhynchus heterochetus* was recovered as sister to all other Nereididae and Namaneridinae and Dendroneridinae forming a grade with respect to the Gymnonereidinae+ *Paraleonnates*+ Nereidinae clade (Fig 3).

The mitochondrial genome order for all the three newly sequenced taxa was the same as one of the two observed gene orders for Nereididae, identified as Group I by Alves et al. [67]. The newly generated mitogenomes of three nereidids, plus those from 23 other species (plus outgroups), resulted in a concatenated sequence alignment of 2,325 sites for the two *rRNA* genes and 3,829 amino acids for the 13 PCGs. The ML tree (log likelihood = -127521.3745) showed a high bootstrap support (>90%) for many clades (Fig 2B), though some key nodes were recovered with lower support. Allowing for the additional terminals used here, the results were largely congruent with those of Alves et al. [67]. Two major clades were found: Clade I was a well-supported Nereidinae, while Clade II, with relatively low support, consisted of species belonging to Dendronereidinae, *Tylorrhynchus*, Namanereidinae, and *Paraleonnates* (Fig 2B). Dendronereidinae was represented by two terminals, *Dendronereis chipolini* Hsueh, 2019 and *Neanthes glandicincta* (Southern, 1921), although based on the shallow genetic distance between the individuals, the latter terminal is apparently a misidentification. The placement of *Pectinereis strickrotti* gen. nov., sp. nov. within Nereidinae was as the poorly supported sister group to a clade comprised of *Alitta*, *Hediste*, *Nectoneanthes*, *Perinereis*, *Platynereis*, and *Pseudonereis* terminals. The newly generated mitogenome for the type species of *Nereis*, *N. pelagica*, formed a well-supported clade with *Nereis zonata* Malmgren, 1867 while *Nectoneanthes oxypoda* was a well-supported sister group to *Alitta succinea* (Fig 2B).

### Novel morphological features

*Pectinereis strickrotti* gen. nov., sp. nov. specimens show unusual non-reproductive and epitokal morphology among nereidids by the presence of five autapomorphic features. Two of them

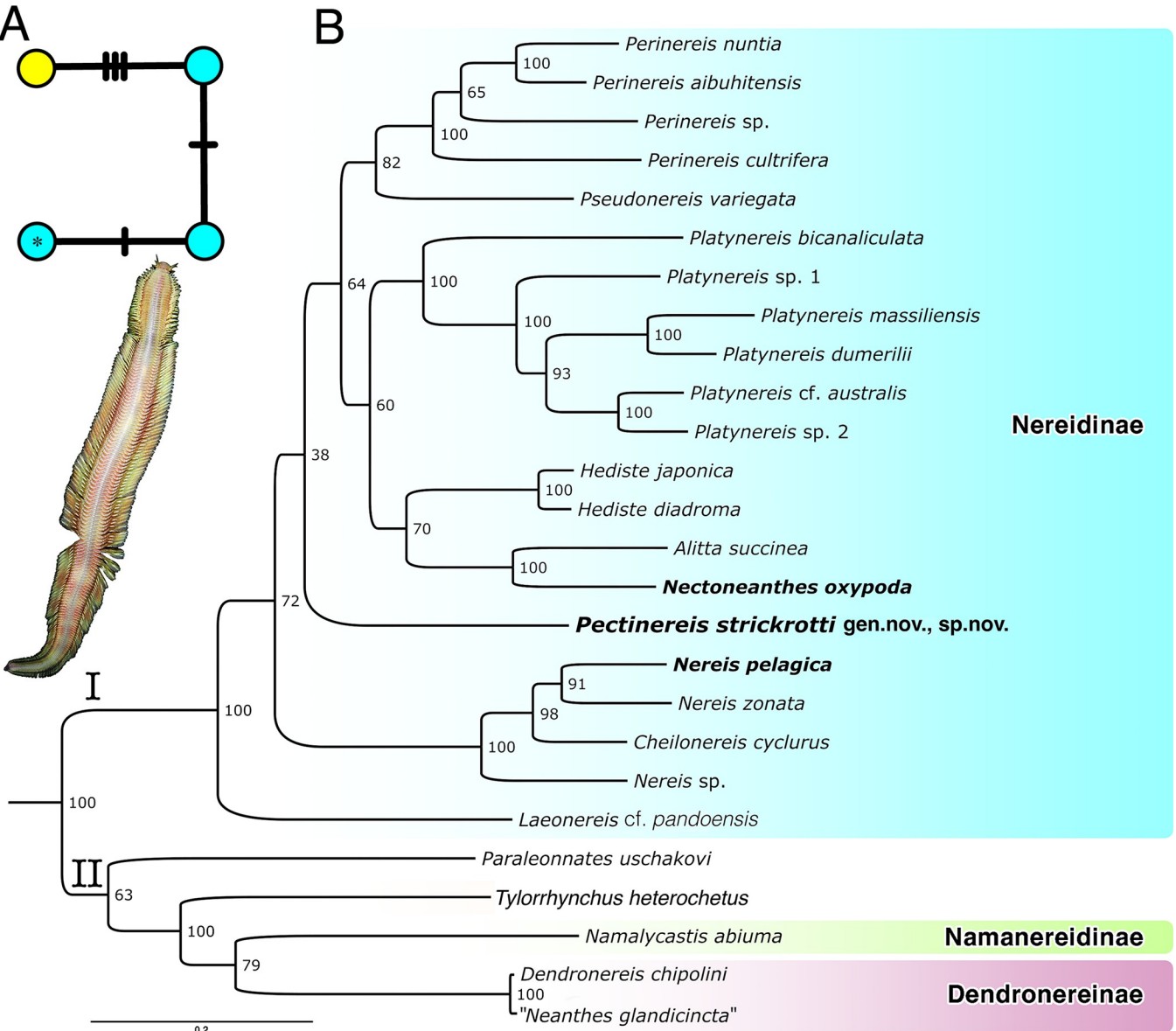

**Fig 2. Haplotype analysis and mitogenome phylogeny.** A. Haplotype network of *COI* data acquired for the three male and one female specimen of *Pectinereis strickrotti* gen. nov., sp. nov. The holotype sequence (male epitoke) is marked with * and has five base pairs different from the female (infaunal fragment). B. Maximum likelihood (ML) tree derived from analysis of the concatenated 15-gene mitochondrial genome dataset, with the 13 PCGs translated to amino acids. Support values at nodes are bootstrap support percentages after 1,000 pseudoreplicates.

unrelated to the reproductive modifications: (A) pectinate branchiferous parapodial cirrostyles, and (B) elongate, fusiform dorsal ligule emerging basally to expanded cirrophores. And the other three developed during males epitoky (unknown in females, see species 'Remarks' below): (C) body divided into four regions, (D) hooked aciculae, and (E) elfin-shoe shaped ventral cirrophores. Each of those diagnostic characters makes the new genus unique within the family, as demonstrated in both the morphological (see below) and the phylogenetic analyses (Figs 2B and 3). Hence, a new genus is established and a new species is described. A detailed comparison between *Pectinereis* gen. nov. and other closely related genera is given in the Remarks section.

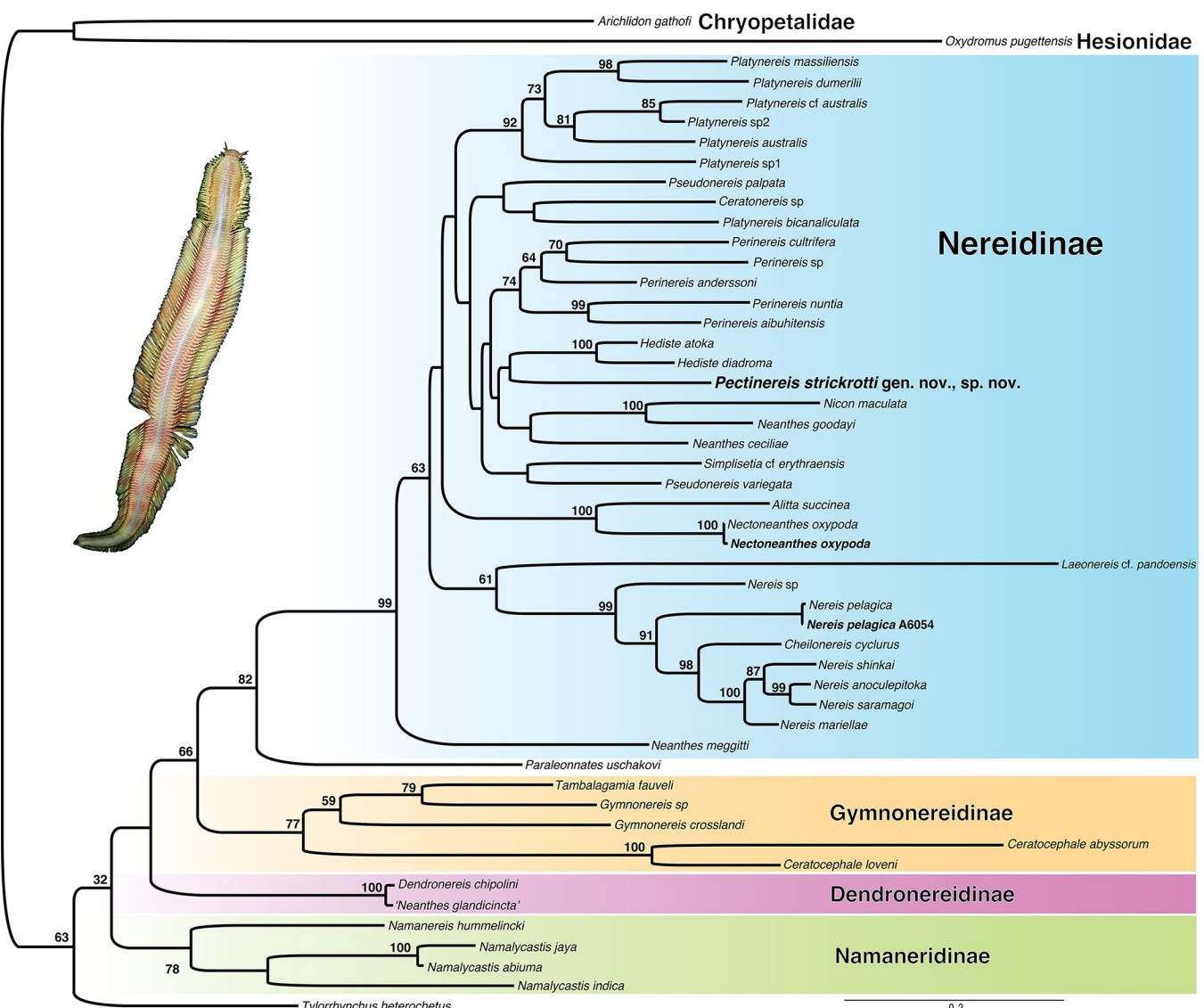

**Fig 3. Maximum likelihood (ML) tree derived from analysis of nuclear 18S rRNA and mitochondrial COI and 16S rRNA DNA sequences.** Support values at nodes are bootstrap support percentages after 1,000 pseudoreplicates.

## Non-reproductive morphology

<u>Gills</u>. Oxygen exchange in nereidids is taken over by the branchiae, which are present as vascularized parapodial structures [1]. Elaborate branchiae, namely gills, develop only in a few genera; they are associated with the dorsal cirrophore and start some distance from the prostomium. The most distinctive and complex gills have been reported in two genera: *Dendronereides* Southern, 1921, and *Dendronereis* Peters, 1854. The gills in *Dendronereides* are arborescent with branched bunches of filaments inserted basally on the cirrophore and above the median ligule. In contrast, those in *Dendronereis* Peters, 1854 are markedly modified dorsal cirrophores with bipinnate shapes, consisting of branches arising laterally from the primary axis. Enlarged dorsal cirrophores have also been reported as gills in *Gymnonereis* [95], *Namalycastis* [89, 96], *Nereis* [97], and *Tambalagamia* Pillai, 1961 [98], and those highly vascularized should also be treated in the same manner in members of *Alitta* (see [49]: Figs 2F and 3E; [99]:

Figs 3E–3G, 9F, 17K and 18G), *Nectoneanthes* ([100]: Figs 8D and 11D and 14D), *Stenoninereis* ([90]: Figs 2D and 5D), as well as some *Neanthes* with enlarged dorsal cirrophores (see [101]: table II, as shape of dorsal ligule), among others.

The gills in *Pectinereis* gen. nov. are an exceptional case among nereidids. These respiratory organs are modified dorsal and ventral cirrostyles of anterior chaetigers, with the former thicker and encompassing a slightly higher range of chaetigers. They are pectinate in shape by the 4–11 filaments that arise from the upper edge and vascularized with a main broad blood vessel that runs along the stem and branches out to the filaments. In contrast to nereidids with gills associated to dorsal cirrophores and starting at some distance from the prostomium, these structures in *Pectinereis* gen. nov. are associated not only with the dorsal but also to the ventral cirrostyles, although with a slightly more restricted distribution in anterior parapodia.

*Pectinereis* gen. nov. gills may remind the shape of scalloped or swollen cirrostyles of some epitokous nereidids, which are associated with the chemical reception of pheromones [102, 103]. Nevertheless, the former are short, knob-like structures exclusive of natatory parapodia present typically in males, whereas those in *Pectinereis* gen. nov. are elongate and digitiform, restricted to the pre-natatory parapodia. Although the swollen cirrostyles of epitokous nereidids and the pectinate cirrostyles of *Pectinereis* gen. nov. are present in both dorsal and ventral cirri of pre-natatory parapodia, both forms are very different by themselves. Additionally, the swollen cirrostyles are present in up to the first 7–8 parapodia, whereas the pectinate cirrostyles of *Pectinereis* gen. nov. are present at least in the first 14 parapodia. These structures are not homologs due to their morphology and function. Therefore, the presence of gills can be considered a non-reproductive modification in *Pectinereis* gen. nov.

Dorsal ligule + dorsal cirrophore. When present in nereidids, the dorsal ligule is attached frontally to the dorsal cirrophore, although it seems to be attached to the notoacicular ligule itself instead of the dorsal cirrophore in *Stenoninereis* species (see [90]). The division between the dorsal ligule and the dorsal cirrophore appears as an intermediate constriction that runs typically from the dorsal cirrostyle towards the base of the dorsal ligule (see [4]: Fig 1C).

When the dorsal cirrophore enlarges progressively, the dorsal ligules change size, location, or both. The following combinations are usually recognizable in nereidids with dorsal cirrophores enlarging towards posterior end: (I) dorsal ligule of similar size throughout body but located distally in posterior parapodia, with base occupying more or less half of dorsal cirrophore, as occurs in some *Neanthes* (see [104]: Pl. 3, Figs 12–14) and *Perinereis* (see [105]: Figs 13 and 16) species; (II) dorsal ligule becoming strongly reduced and located distally in posterior parapodia, with base occupying a small part of dorsal cirrophore, as occurs in *Ceratonereis* (see [106]: 4M, 6H), *Pseudonereis* (see [107]: 9K; [108]: 2E, F) or *Alitta succinea* group species (see [49]: Fig 2E and 2F); and (III) dorsal ligule becoming enlarged, with base occupying all the frontal flank of dorsal cirrophore, as shown in *Cheilonereis* (see [109]: Fig 6F and 6G), *Nectoneanthes* (see [100]: Figs 11C and 13D), and *Alitta virens* group (see [99]: Fig 7G and 17K) species. However, an additional form can be recognized solely in *Pectinereis* gen. nov. The (IV) dorsal ligule is of similar size throughout the body but located sub-medially in posterior chaetigers, its base occupies a small part of the dorsal cirrophore, giving a false impression of the cirrophore being 'bifurcated.' This fourth combination of dorsal ligule + dorsal cirrophore ligule also makes the new genus unique among nereidids.

Epitokal morphology. Epitokous nereidids are divided in two (pre-natatory and natatory) or three (also post-natatory) body regions, whose reproductive morphological features and their function have been largely addressed in much detail in the literature [1, 5, 6, 110–112]. Interestingly, the epitokes of *Pectinereis* gen. nov. have body divided into four regions: pre-natatory, natatory, post-natatory, and pre-pygidial. During epitoky, chaetigers of the posterior end show less modification than medial segments and are the last to start a transformation

process, if at all, because they may present slight changes or remain unmodified [112, 113]. This also occurs with the post-natatory chaetigers of *Pectinereis* gen. nov.; parapodial ligules become shorter and cirrostyles elongated posteriorly. However, another distinct body region can be seen between post-natatory chaetigers and pygidium, here referred to as pre-pygidial. Chaetigers of this region are evidently narrower than post-natatory ones, with most of the parapodial projections markedly reduced, barely noticeable, and the dorsal cirrostyles were all detached. Also, the ventral ligules are short and the ventral cirrostyles very much more elongated than in previous chaetigers; however, the most strikingly novel feature of this region is the presence of hooked aciculae (see below). *Pectinereis* gen. nov. is the only nereidid with a fourth body epitokal region, and it makes the genus unique within the family.

Aciculae. These supportive chaetae in nereidids are deeply embedded within the parapodia so that only sometimes its small tip emerges from the body surface. The typical aciculae have a billiard cue shape—gradually tapering towards the distal end, straight, slender, with a truncate proximal end—sometimes slightly curved, particularly in epitokous parapodia; however, in mature *Tambalagamia fauveli* Pillai, 1961 they are sharply curved at the tip with a marked sickle shape in natatory chaetigers [6].

Remarkably, some of the aciculae in *Pectinereis* gen. nov. differ from all the forms previously recorded in nereidids. Although the aciculae present are mainly of the typical form, those shown in the notopodia and neuropodia of most posterior chaetigers (pre-pygidial region), located just immediately before the pygidial rosette—a sex-specific epitokal structure for releasing the sperm through developed papillae [112, 114, 115]—have a more robust and stouter appearance with a falcate distal end, exhibit a curved body that tapers towards a blunt proximal end, and protrude conspicuously beyond the parapodial surface. All these features of the 'hooked aciculae' are generally more pronounced in the notopodia. They seem to resemble the sickle-shaped aciculae of *T. fauveli* epitokes, although they differ in form and appearance on the body. For instance, the hooked aciculae in *Pectinereis* gen. nov. in the post-natatory parapodia are stouter and curved with a falcate distal end, whereas the sickle-shaped aciculae of *T. fauveli* are slender, straight, with a sharply curved tip, and restricted to the natatory parapodia.

*Pectinereis* gen. nov. hooked aciculae presumably develop during male sexual maturation before spawning. The appearance of specialized chaetae within epitokous nereidids is a frequent phenomenon. It generally involves the emergence of specific natatory compound chaetae to facilitate swimming within the water column for swarming, which depending on the species, can be paddle-like [112, 115, 116], capillaries [117] or ensiform [99]. However, quite different chaetae have rarely been reported within nereidids at maturity. These are hook-shaped non-acicular chaetae ornamented with dorsal spines and forming part of the neuropodial bundle of the third chaetiger in some *Micronereis* species (see [118]: Figs 9, 25 and 30]. These specialized chaetae have been suggested as male genital structures with supporting copulatory functions that are probably used to pierce the epidermis of the posterior dorsal surface of the female for an eventual sperm transfer [118–120]. Analogous chaetae have been found mainly in the meiofaunal syllid *Sphaerosyllis hermaphrodita* Westheide, 1990, where the aciculae of a single chaetiger are modified as a solid and straight copulatory structure with a curved and blunt distal end, subdistally broadened and flattened, and ornamented with teeth [121, 122].

The function of the hooked pre-pygidial aciculae in *Pectinereis* gen. nov. is uncertain. Their location and form in the males of *P. strickrotti* gen. nov., sp. nov., as well as the appearance of modified chaetae in epitokous males of a few nereidids and syllids during reproduction, might suggest some hints on the hooked aciculae function. Epitoke males might produce mechanical body wall ruptures on the fully mature female through the hooked aciculae to discharge the

sperm immediately after using the pygidial papillae. Although it is likely that females remain atokous dwelling in the sediment (see below, 'Biology'), successful reproduction in nereidids needs close interaction or contact from a partner of the opposite sex to release the genital products given a chemical stimulus [119, 123–127]. When this approach occurs between reproductive individuals of *P. strickrotti* gen. nov., sp. nov. is unknown. The reliable purpose of hooked pre-pygidial aciculae has yet to be discovered. A detailed study of the reproductive behavior of this species is required.

<u>Ventral cirrophores</u>. Nereidids have ventral cirri consisting of two main components: (1) the distal, usually elongated cirrostyle; and (2) the proximal, slightly or barely developed cirrophore (rarely undeveloped as in *Micronereis* species, see [118]). These ventral poorly-developed cirrophores are barely noticeable in atokous nereidids or non-natatory regions of epitokes. However, in natatory parapodia of epitokous nereidids, they are typically well-developed as an enlarged and highly vascularized membrane divided into upper and lower lamellae. The upper lamella is generally less developed and may present an additional secondary flap, unlike the single and reniform foliose lower lamellae.

In the pre-natatory parapodia of *Pectinereis* gen. nov., the ventral cirrophores show the typical non-modified form. Interestingly, in the natatory chaetigers, they are notably different from other epitokous nereidids. The ventral cirrophores are markedly elongated and acuminate, with only a drop-shaped and flattened lower lamella, whose overall form is reminiscent of an elf's shoe. This has not previously been documented within Nereididae. The elfin-shoe-shaped cirrophore is assumed here to be an epitokal modification probably used as an oar to move forward during the swimming behavior of males.

## Taxonomy

Family NEREIDIDAE de Blainville, 1818
  Subfamily NEREIDINAE de Blainville, 1818

## *Pectinereis*

Villalobos-Guerrero, Huč, Tilic, Hiley & Rouse **gen. nov.**
  urn:lsid:zoobank.org:act:17ECB80D-BCC4-42F0-9F4E-3DD42EEF5EAA
  Type species. *Pectinereis strickrotti* Villalobos-Guerrero, Huc, Tilic, Hiley & Rouse sp. nov.

## Diagnosis

Prostomial anterior region entire. Esophageal caeca absent. Anterior parapodial cirrostyles as comb-like gills. Dorsal cirrostyles attached sub-distally and dorsal ligule attached sub-medially to expanded cirrophores. Notopodial prechaetal, neuropodial postchaetal and inferior lobes present. First two chaetigers without notoacicula. Neuropodial spinigers and falcigers very long, homogomph. Epitoke males divided into four body regions, with distally-bilamellated dorsal cirrophore, elfin-shoe shaped ventral cirrophore, pre-pygidial hooked aciculae, and ensiform spinigers.

## Description

Prostomium with anterior region entire; longitudinal groove present. Paired antennae present. Palpophores sub-conical, oriented downwards, with conspicuous transverse groove; palpostyles digitiform. Eyes and lens absent. Anterior achaetous segmental region (tentacular belt) without ventrolateral projections, bearing four pairs of enlarged (tentacular) cirri. Proboscis with cylindrical rings; paragnaths only, conical, evenly spaced, present on both rings. Paired

esophageal caeca absent. Segmental glandular patches present. Notopodia well developed from third chaetiger. Parapodial (dorsal and ventral) cirrostyles as comb-like gills in first anterior chaetigers, with digitiform filaments, smooth in following chaetigers. Dorsal cirrostyle attached sub-distally to expanded cirrophores. Dorsal cirrophore markedly enlarged in mid-body and posterior chaetigers; two divergent vessels running lengthwise. Dorsal, median, and ventral ligules present. Dorsal ligule elongate, fusiform, attached sub-medially to enlarged cirrophores. Notopodial prechaetal lobe present throughout body. Median and ventral ligule smooth, fusiform. Neuropodial postchaetal lobe present throughout body. Neuropodial superior lobe absent. Neuropodial inferior lobe present, restricted to a few anterior chaetigers. Ventral cirrostyle single. Aciculae mostly dark throughout. Notoaciculae absent in first two chaetigers. Notochaetae with homogomph spinigers. Neurochaetae of both fascicles with homogomph spinigers and homogomph falcigers. Blade of falcigers very long, distal end obliquely truncate, terminal tooth without loop.

Epitokous males with body divided into four regions: pre-natatory, natatory, post-natatory, and pre-pygidial. Dorsal cirrophores distally bilamellated (upper and lower lamellae) in natatory chaetigers. Ventral cirrophore elfin-shoe shaped (markedly elongated, acuminated, with lower lamella only) in natatory chaetigers. Aciculae hook-shaped in pre-pygidial chaetigers. Notoaciculae without expanded basal end. Sesquigomph epitoke spiniger ensiform, present in notopodia and both fascicles of neuropodia.

## Etymology

This genus is named by combining the Latin word *pectinis* (= 'comb') with the name of the type genus of the family, *Nereis*. The name emphasizes the pectinate (i.e., comb-like) parapodial cirrostyles (gills) in the first anterior chaetigers formed by digitiform filaments. The gender is feminine, as the stem genus-group name.

## Remarks

*Pectinereis* gen. nov. sits well within the subfamily Nereidinae as earlier delineated by Fitzhugh [128] and currently conceived by Alves and colleagues [129]. Gills in nereidids are vascularized parapodia structures that have been recorded only in two shallow-water and estuarine genera: *Dendronereis* Peters, 1854 and *Dendronereides* Southern, 1921. The gills are modified dorsal cirrophores with multiple filaments starting at least some chaetigers after the first one. However, in the deep-water *Pectinereis* gen. nov., the gills are modified dorsal and ventral cirrostyles present from the first chaetiger to a few anterior ones. In addition, *Pectinereis* gen. nov. can readily be distinguished from *Dendronereis* and *Dendronereides* by having an anteriorly complete prostomium, two neuropodial (postchaetal and inferior) lobes, and paragnaths only on pharyngeal rings, whereas those two genera have an anteriorly indented prostomium, at least three neuropodial lobes, and papillae only on pharyngeal rings, when present.

In nereidids, the ventral cirrophore is poorly developed compared to the dorsal cirrophore, and this is possibly the reason for having overlooked it in the family's systematics. In some epitokes, however, it is enlarged with additional lamellae but still not distinguished in literature from the cirrostyle, referring to it usually as 'ventral cirrus'. *Pectinereis* gen. nov. is unique in that it shows an elfin-shoe shaped ventral cirrus in natatory chaetigers, viz., a markedly elongated ventral cirrophore with a lower and drop-shaped lamellae, becoming distally acuminated, where the ventral cirrostyle is attached. Epitoke individuals from other nereidid genera show a short and cylindrical ventral cirrophore in the natatory chaetigers with a reniform lower lamella and, at least, one digitiform upper lamella.

The markedly enlarged dorsal cirrophore is present in the medial and posterior chaetigers of several nereidid genera, whether or not they are in an epitoke stage. For instance, *Alitta* Kinberg, 1865, *Cheilonereis* Benham, 1916, *Neanthes* Kinberg, 1865, *Nectoneanthes* Imajima, 1972, *Nereis* Linnaeus, 1758, *Paraleonnates* Khlebovich & Wu, 1962, *Perinereis* Kinberg, 1865, and *Pseudonereis* Kinberg, 1865. Nonetheless, *Pectinereis* gen. nov. can be distinguished from all those genera because the base of the elongate and fusiform dorsal ligule is attached sub-medially to the enlarged dorsal cirrophore, giving the false appearance of a bifurcate cirrophore. In contrast, the '*A. succinea*' species complex and some *Neanthes*, *Nereis*, *Perinereis*, and *Pseudonereis* species, the dorsal ligule is smaller and shifted toward the distal end of the enlarged dorsal cirrophore, although sometimes it is completely reduced in the posterior chaetigers. On the other hand, in the '*A. virens*' species complex, *Cheilonereis*, *Nectoneanthes*, and some *Paraleonnates* species the dorsal ligule is broadly enlarged and embraces entirely the enlarged dorsal cirrophore. *Typhlonereis* Hansen, 1879 was until now the single nereidid genus exclusive from the deep sea. Over 140 years later, *Pectinereis* gen. nov. is established as endemic to deep environments.

## Pectinereis strickrotti

Villalobos-Guerrero, Huč, Tilic, Hiley & Rouse **sp. nov.**

Figs 1 and 4–8

urn:lsid:zoobank.org:act:E143D2DF-BDD4-4E61-B5E7-5218EDD6B2A5

## Material examined

*Holotype*. SIO-BIC A9836, epitoke male, Mound 12, Costa Rica, Pacific Ocean (8.929° N; 84.313° W), 02 Nov. 2018, 1,001–1,010 m, dive AD4987 (black slurp), coll. E. Cordes, E. Cowell, R/V *Atlantis*, DSV *Alvin*, fixed in 95% EtOH, swimming near the bottom, in good condition. GenBank *COI* sequence OQ415952, mitochondrial genome OL782600, 18S OR437941.

*Paratypes*. One epitoke male (MZUCR XXXX, was SIO-BIC 9837), same data as holotype, good condition, GenBank *COI* sequence OQ415953; one epitoke male (SIO-BIC A9889), Mound 12, Costa Rica, 30 Oct. 2018, 997 m, dive AD4984 (red slurp), coll. S. Goffredi, O. Pereira, R/V *Atlantis*, DSV *Alvin*, fixed in 10% formalin, swimming near the bottom, in good condition. GenBank *COI* sequence OQ415954; one incomplete female, mid-body only (SIO-BIC A9891), Mound 12, Costa Rica, 30 Oct. 2018, 996 m, dive AD4984, coll. S. Goffredi, O. Pereira, R/V *Atlantis*, DSV *Alvin*, sediment of pushcore, fixed in 10% formalin. GenBank *COI* sequence OQ415955.

*Description*, *holotype epitoke male*. Incomplete, 69 mm LT, 13 mm L15, 5 mm W15, with 150 chaetigers (paratype SIO-BIC A9889 complete with 192 chaetigers). General body color yellowish to reddish with mid-dorsum of chaetigers gray in live specimens, without pigmentation patterns but marked iridescence throughout (Fig 4A and 4B); body cream (Fig 4C) with faint traces of brownish pigmentation in palpophore and palpostyles (Fig 5A), and single transverse row of same color on dorsum of preserved specimens, more enhanced on posterior chaetigers.

Prostomium pear-shaped (Figs 4C, 4D and 5A), bent downwards in paratypes with non-everted proboscis, division between regions barely seen; anterior region distally entire, sub-rounded, as long as posterior region; anterolateral gap between antenna and palpophore narrow, as wide as basal diameter of antennae (Fig 4D). Nuchal organs covered by achaetous anterior segmental region (tentacular belt) bearing enlarged cirri, deeply embedded, medium size, as wide as basal diameter of posterodorsal enlarged anterior cirri.

Palpophores sub-conical, slightly thick, as long as wide (Figs 4D, 5A and 5C), as long as prostomium, bent downwards in paratypes with non-everted proboscis; sub-distal transverse

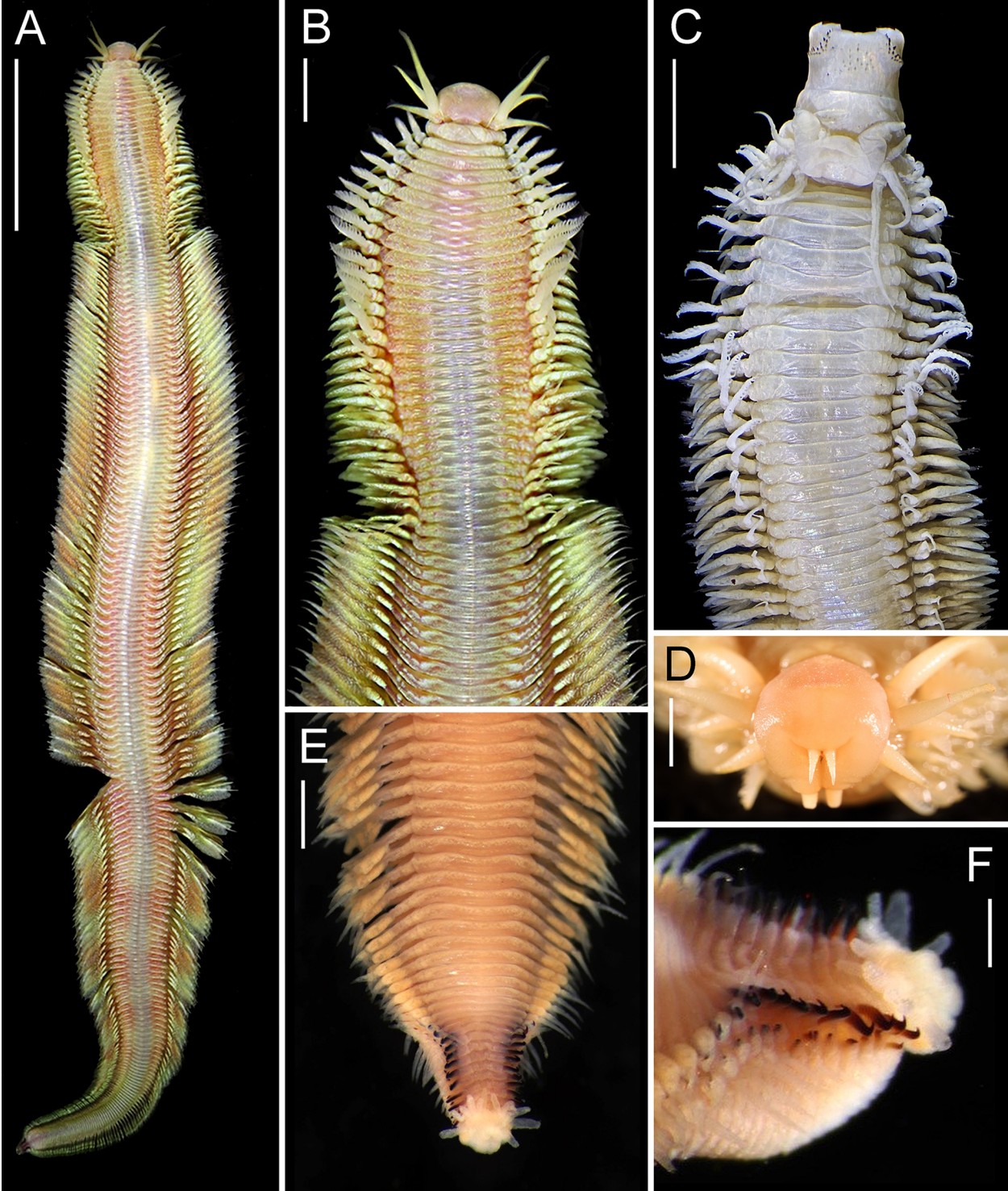

**Fig 4. *Pectinereis strickrotti* gen. nov., sp. nov. male anatomy.** A, B, D-F. Paratype (SIO-BIC A9889), epitokous male: A, whole body of living specimen in dorsal view; B, anterior region of living specimen in dorsal view; D, prostomium of preserved specimen in frontal view; E, posterior end of preserved specimen in dorsal view; F, post-natatory chaetigers and pygidium of preserved specimen in dorsolateral view. C. Holotype (SIO-BIC A9836), epitokous male: anterior region of preserved specimen in dorsal view. Scale bars: A, ~20 mm; B, ~5 mm; C, 5 mm; D, 1 mm; E, 3 mm; F, 0.5 mm. Credits: A, B, Ekin Tilic; C, Tulio Villalobos; D-F, Greg Rouse.

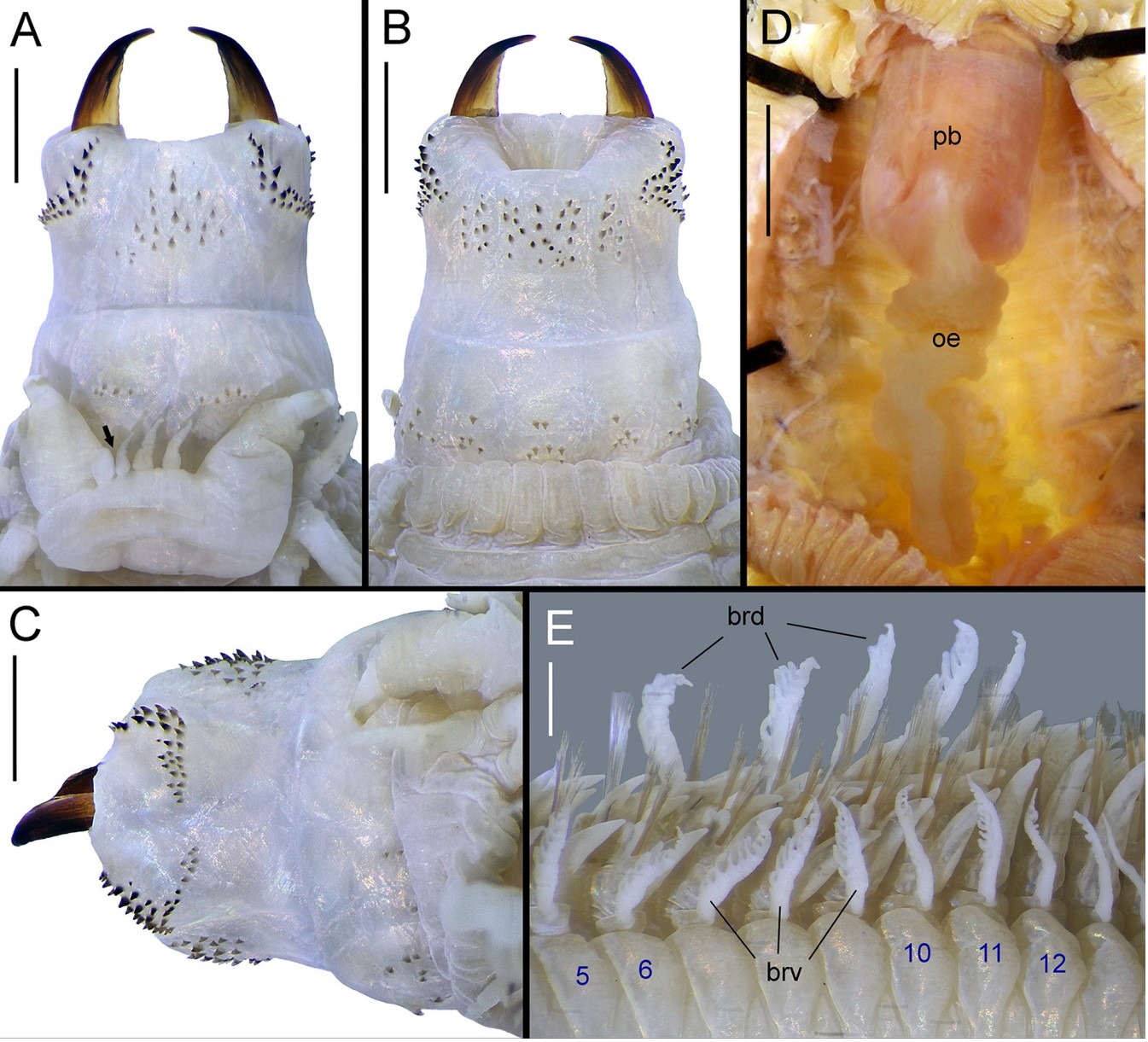

**Fig 5. *Pectinereis strickrotti* gen. nov., sp. Nov. male anatomy details.** A, B, D, E. Holotype (SIO-BIC A9836), epitokous male: A, prostomium and everted proboscis in dorsal view (arrow pointing additional abnormal antennae); B, everted proboscis in ventral view; C, everted proboscis in lateral view; E, branchiferous chaetigers in ventrolateral view (numbers referring to chaetiger). D. Paratype (SIO-BIC A9889), epitokous male: dissected anterior end in ventral view. Abbreviations: brd, dorsal gills; brv, ventral gills; pb, pharyngeal bulb; oe, esophagus. Scale bars: A–C, 2 mm; D, 3 mm; E, 1 mm. Credits: A-C, E: Tulio Villalobos; D, Greg Rouse.

groove distinct (Fig 5A). Palpostyles oval, as wide as one-quarter of palpophore (Figs 4D and 5A). Antennae abnormally as two pairs (Fig 5A), one pair only in paratypes; tapered, conical, medium size, as long as one-half of prostomial posterior region; antennae slightly separated by gap as wide as one-third of basal diameter of antenna (Figs 4D and 5A). Eyes and lens absent (Figs 4D and 5A).

Anterior achaetous segmental region (tentacular belt) markedly contracted owing to fixation and protrusion of proboscis, with straight anterior margin, bearing enlarged anterior cirri. Enlarged anterior cirri with rugged, non-articulated cirrostyles (Fig 4C and 4D).

Anterodorsal cirrostyles extending backwards to chaetiger 4 (3 in paratype). Anteroventral cirrostyles longer than palpophores, as long and thick as posteroventral cirrostyles. Posterodorsal cirrostyles longest, extending backwards to chaetiger 8 (9 in paratype). Posteroventral cirrostyles extended laterally over middle of prostomial posterior region. Dorsal and posteroventral cirrophores cylindrical, anteroventral cirrophores ring-shaped; anteroventral cirrophores as wide as posteroventral cirrophores.

Proboscis everted, with maxillary and oral rings cylindrical, wider than long (Fig 5A–5C). Proboscis structures observed in holotype only. Jaws slightly crenulate (Fig 5A and 5B), reddish in distal third, remaining amber; with faint traces of 5 short denticles; 2 canals emerging from pulp cavity.

Paragnaths present on both maxillary and oral rings of proboscis, all conical, dark red and brown in maxillary ring, brownish in oral ring (Fig 5A–5C); plate-like basements absent. Area I: 18, five slightly regular rows of uneven cones in broad, triangular patch, medial row cones largest (Fig 5A and 5C). Areas IIa: 26, IIb: 29, two to three regular longitudinal rows of uneven cones in L-shaped patch, distal cones larger (Fig 5A and 5C). Area III: 42, five irregular rows of uneven cones in broad, rectangular patch; proximal and outer cones shorter; 9 and 11 laterally-isolated cones in three slightly regular longitudinal rows (Fig 5B and 5C). Areas IVa: 33 and IVb: 32, three slightly curved longitudinal rows of uneven cones in L-shaped patch, distal cones larger forming one regular, transverse row (Fig 5B and 5C). Area V: 0 (Fig 5A). Areas VIa: 5 and VIb: 5, one slightly regular, transverse row of even cones (Fig 5A). Areas VII–VIII: 47, two well-separated bands of even cones on ridges only (absent in furrows), as isolated irregular patches; anterior band consisting of one regular transverse row of six cones (two on area VII, one on each ventral ridge of area VIII); posterior band with two transverse irregular rows (4–5 on each ventral ridge) (Fig 5B and 5C). Ridges of areas VI–V–VI with λ-shaped pattern (Fig 5A). Gap between area VI and areas VII–VIII broad, as wide as distal end of palpophore. Paired esophageal caeca absent (Fig 5D).

Body incomplete, divided into two regions although lacking posterior end, complete paratypes with four regions (Fig 4A): Holotype consisting of pre-natatory region with 31 chaetigers (Fig 4C) and natatory region with 119 chaetigers, becoming gradually narrower towards posterior end from about chaetiger 125; in paratypes, pre-natatory region with 30–31 chaetigers (Fig 4A and 4B), natatory region with 94 chaetigers (Fig 4A), post-natatory region with 54 chaetigers (Fig 4A and 4E), and pre-pygidial region with 13 chaetigers (Fig 4E and 4F).

Pre-natatory region (Fig 6A–6E) with notopodia consisting of dorsal cirri with cirrostyle and cirrophore, dorsal ligule, notopodial prechaetal lobe, and median ligule in biramous parapodia; and neuropodia consisting of neuroacicular ligule with inferior and postchaetal lobes, ventral ligule, and ventral cirrus with cirrostyle and cirrophore (neuropodial superior lobe not developed). First 17–18 dorsal and 14 ventral cirrostyles markedly modified as pectinate gills (Figs 4B, 6C, 5E and 6A–6D): dorsal gills thick, becoming narrower and extending beyond dorsal ligule in parapodia 1–13 (Fig 5E), reducing in size progressively to become as long as dorsal ligule in chaetigers 17–18; ventral gills narrower than dorsal ones, becoming slenderer and extending beyond ventral ligule in chaetigers 1–14 (Fig 5E). Dorsal and ventral gills with filaments on upper edge (Figs 5E, 6A–6D); dorsal branchial filaments becoming shorter and narrower toward distal end of gill, 6–11 filaments, more abundant in parapodia 3–14; ventral branchial filaments becoming longer and thicker toward middle of gill, 2–10 filaments, more abundant in parapodia 4–10 and decreasing drastically in following parapodia with gills (Fig 7). Dorsal cirrostyles of parapodia 18–19 to 31 and ventral cirrostyles of parapodia 15–31 cirriform, smooth (Fig 6E); dorsal and ventral cirrostyles elongating gradually to become as long as dorsal ligule and ventral ligule, respectively. Dorsal and ventral cirrophores enlarging from chaetigers 28 and 19, respectively. Parapodial ligules long, slender, tapering; dorsal ligule

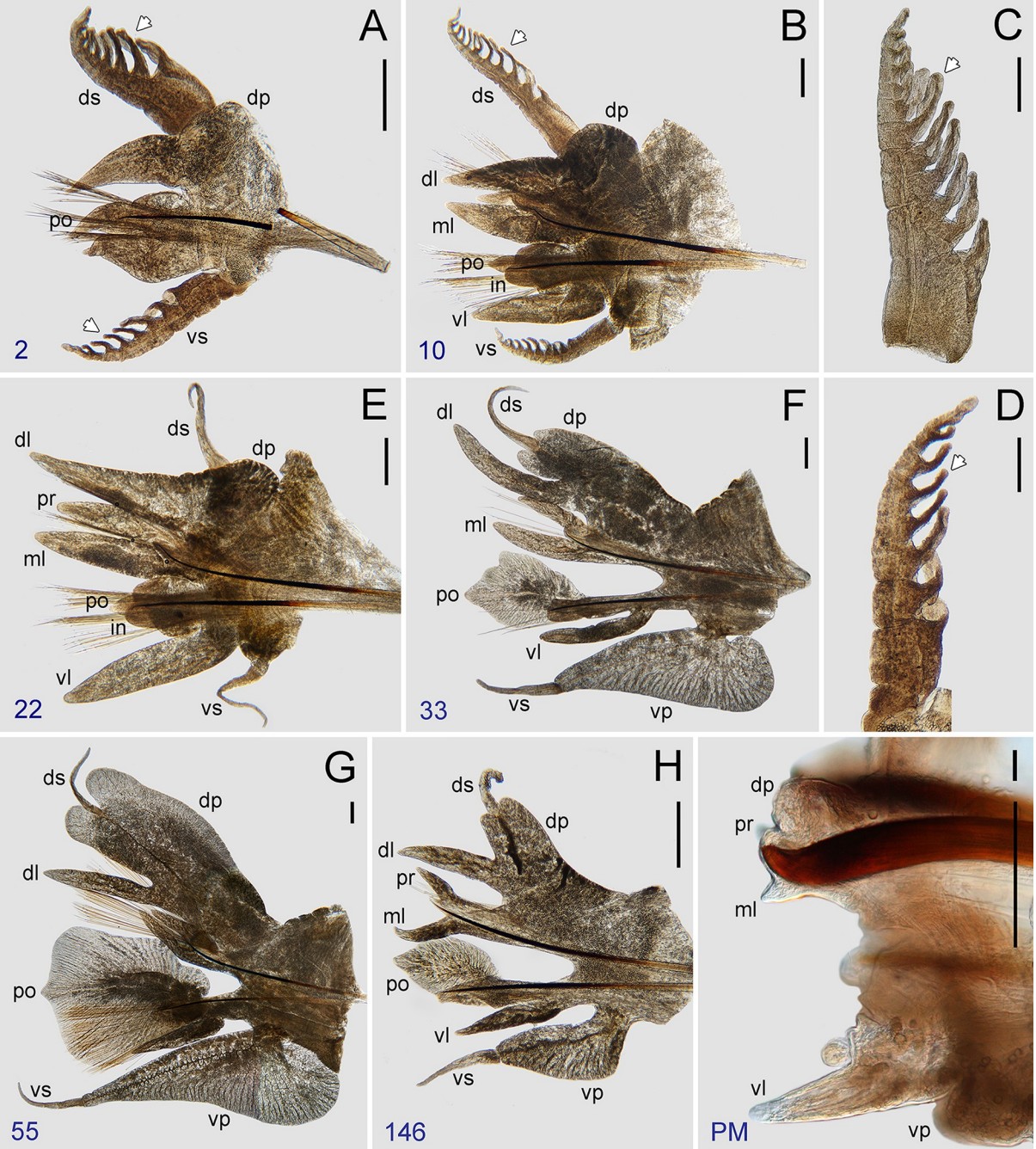

**Fig 6. *Pectinereis strickrotti* gen. nov., sp. nov. male parapodia.** A-H. Holotype (SIO-BIC A9836), epitokous male: A, parapodium of chaetiger 2; B, parapodium of chaetiger 10; C, parapodial dorsal cirrus of chaetiger 5; D, parapodial ventral cirrus of chaetiger 5; E, parapodium of chaetiger 22; F, parapodium of chaetiger 33; G, parapodium of chaetiger 55; H, parapodium of chaetiger 146. I. Paratype (SIO-BIC A9889), epitokous male: parapodium of pre-pygidial (PM) chaetigers. White arrows indicate branchial filament. Abbreviations: dl, dorsal ligule; dp, dorsal cirrophore; ds, dorsal cirrostyle; in, inferior lobe; ml, median ligule; po, neuropodial postchaetal lobe; pr, notopodial prechaetal lobe; vl, ventral ligule; vp, ventral cirrophore; vs, ventral cirrostyle. Scale bars: A, B, E-H, 0.5 mm; C, D, I, 0.2 mm. Credits: A-H, Tulio F. Villalobos; I, Greg Rouse.

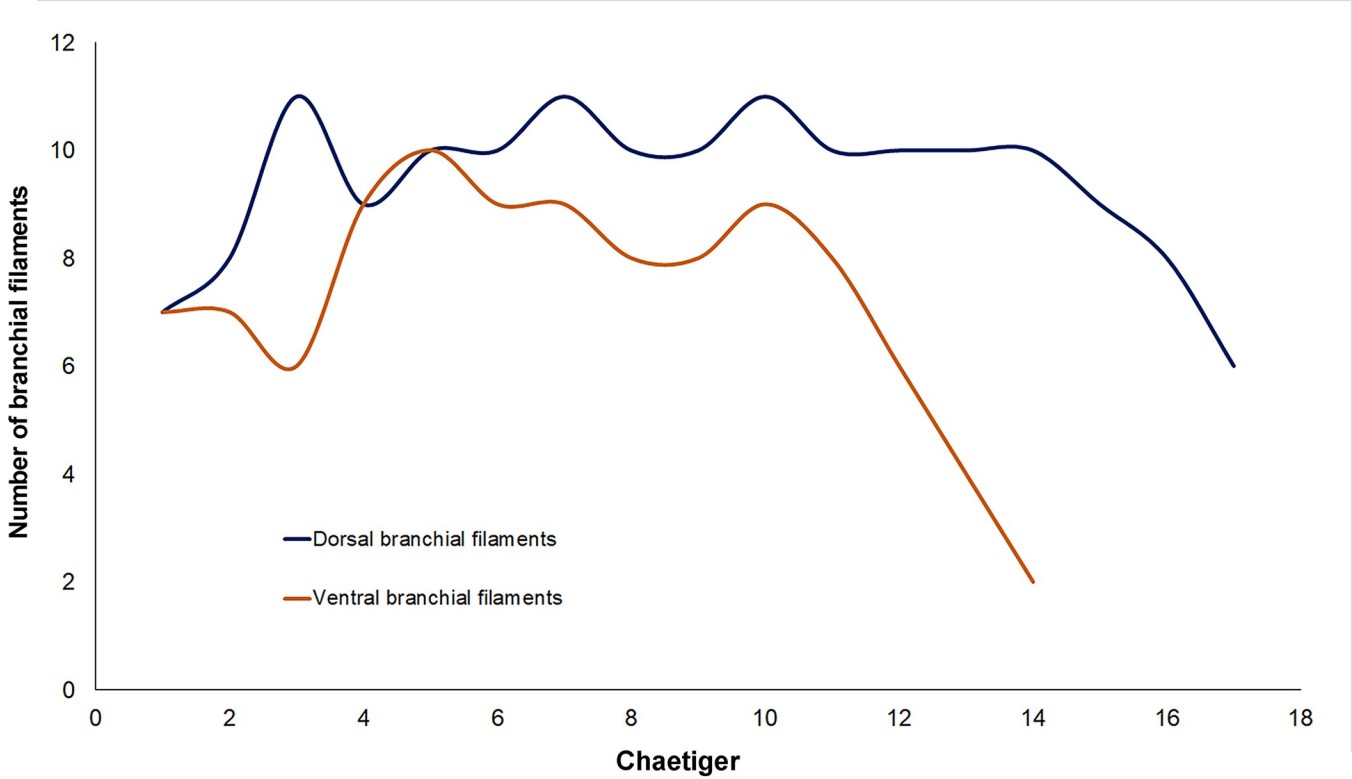

**Fig 7. Distribution of dorsal and ventral branchial filaments of right parapodia from the holotype (SIO-BIC A9836) of *Pectinereis strickrotti* gen. nov., sp. nov.**

sharper, subequal to median ligule (Fig 6A, 6B and 6E). Notopodial prechaetal lobe present from chaetiger 3, long, slender, tapering, as long as three-quarters or four-fifths of median ligule (Fig 6A, 6B and 6E). Neuropodial postchaetal lobe conical, slightly longer than neuroacicular ligule (Fig 6A). Inferior lobe slender, conical, longer than neuroacicular ligule in parapodia 1–4 (Fig 6A), blunt and subequal to neuroacicular ligule in following parapodia (Fig 6B and 6E). Neuroacicular ligule shorter than ventral ligule (Fig 6A, 6B and 6E).

Natatory region (Fig 6F–6H) with notopodia consisting of dorsal cirri with cirrostyle and cirrophore, dorsal ligule, notopodial prechaetal lobe, and median ligule; neuropodia consisting of neuroacicular ligule with inferior and postchaetal lobes, ventral ligule, ventral cirri with cirrostyle and cirrophore (neuropodial superior not developed). Dorsal and ventral cirrostyles cirriform, smooth (without papillae), elongated (Fig 6F–6H); dorsal cirrostyle extending beyond dorsal ligule, as long as two-thirds of dorsal cirrophore in anterior natatory chaetigers (Fig 6F), becoming markedly shorter in medial natatory chaetigers (up to one-third of dorsal cirrophore; Fig 6G and 6H), subequal in posterior natatory chaetigers (Fig 6H); ventral cirrostyle extending markedly beyond ventral ligule, as long as two-fifths to one-third of dorsal cirrophore in anterior and medial natatory chaetigers (Fig 6F–6G), as long as three-quarters in posterior natatory chaetigers (Fig 6H). Dorsal cirrophore expanded, sub-rectangular, distally bilamellated with subequal, tongue-shaped upper and lower lamellae, more distinct in medial natatory chaetigers; two divergent vessels running lengthwise (Fig 6G). Dorsal ligule fusiform, elongated, emerging basally from one-quarter to one-third of lower edge of dorsal cirrophore in all natatory chaetigers (Fig 6F–6H); dorsal ligule extending markedly beyond median ligule in anterior and medial natatory chaetigers, subequal in posterior natatory chaetigers.

Notopodial prechaetal lobe and median ligule non-modified, slender, tapering, nearly subequal but lobe shorter (Fig 6F–6H). Neuroacicular ligule elongated, slender, markedly shorter than ventral ligule. Neuropodial postchaetal lobe with upper lamella from parapodia 32, broadly rhomboid in parapodia 32–35 (Fig 6F), markedly expanded, asymmetrical, cordiform with distal tip in medial natatory chaetigers (Fig 6G), shorter and leaf-like in posterior natatory chaetigers (Fig 6H). Inferior lobe slightly enlarged in medial natatory chaetigers. Ventral ligule non-modified, slender, tapering, longer than neuroacicular ligule in chaetigers in anterior and medial natatory chaetigers (Fig 6F and 6G), shorter in posterior natatory chaetigers (Fig 6H). Ventral cirrophore markedly elongated, acuminate, with drop-shaped lower lobe only (elfin-shoe shaped; Fig 6F–6H), more distinct in anterior and medial natatory chaetigers.

Post-natatory region lost in holotype, paratype (Fig 4E) with unmodified notopodia, consisting of dorsal cirri with cirrostyle and cirrophore, dorsal ligule, notopodial prechaetal lobe, and median ligule; neuropodia consisting of neuroacicular ligule with postchaetal lobes, ventral ligule, and ventral cirri with cirrostyle and cirrophore (neuropodial inferior and superior lobes not developed). Dorsal and ventral cirrostyles cirriform, smooth; dorsal cirrostyle extending slightly beyond dorsal ligule, as long as dorsal cirrophore, becoming posteriorly markedly longer (up to 2–3 times length of dorsal cirrophore); ventral cirrostyle short, extending to base of ventral ligule, becoming posteriorly longer. Dorsal and ventral cirrophores becoming reduced posteriorly. Parapodial ligules long, slender, tapering; dorsal ligule longer than median ligule. Notopodial prechaetal lobe short, slender, tapering. Neuropodial postchaetal lobe conical, longer than neuroacicular ligule. Neuroacicular ligule slightly longer than ventral ligule.

Pre-pygidial region lost in holotype, paratype (Figs 4E, 4F and 6I) with chaetigers one-third to two-fifths as width as post-natatory chaetigers. Parapodial projections markedly reduced, including dorsal cirrophore, dorsal ligule, and median ligule, and lacking dorsal cirrostyle; ventral ligule short, conical; ventral cirrostyle longest, very elongated, becoming shorter towards pygidium.

Pygidium lost in holotype, metamorphosed in complete paratype (Fig 4E and 4F). Pygidial rosette with 12 dorsal papillae, long, digitiform, subdistal (Fig 4F); anal cirri not present.

Aciculae mostly dark red; present throughout body except notoaciculae absent in first two chaetigers (Fig 6A); notoaciculae with proximal half as wide as neuroaciculae. Aciculae of two types: needle-shaped (Fig 6A, 6B and 6E–6H) and hooked (Figs 4F, 6I, 8B and 8C). Needle-shaped aciculae present almost throughout except most posteriorly, with basal end barely expanded in natatory chaetigers (Fig 8A). Hooked aciculae coarse, protruding distinctly from body surface, with distal end directed upwards (Fig 6I), present only in pre-pygidial chaetigers (Fig 8B and 8C); notoaciculae stouter and more curved (Fig 6B) than neuroaciculae (Fig 6C).

Notochaetae consisting of homogomph spinigers and ensiform chaetae (Fig 8E); homogomph spinigers present in all pre-natatory chaetigers. Upper and lower neurochaetae consisting of homogomph spinigers, homogomph falcigers, and ensiform chaetae; both spinigers and falcigers present in all pre-natatory chaetigers. Ensiform epitokous chaetae replacing all atoke chaetae in notopodia and neuropodia in all natatory chaetigers from chaetiger 32.

Blades of homogomph spinigers finely serrated, with teeth evenly spaced; long, with high b/a ratio. Blades of homogomph falcigers markedly long (b/a ratio: 6.8–7.5; Fig 8D), tip falcate, blunt, without incurved terminal tooth; blade entirely serrated, with slender serrations, pointing upwards, with sub-distal serrations growing laterally to falcate tooth.

## Etymology

The species is named in honor of Bruce Strickrott, Group Manager and lead submersible pilot of the DSV *Alvin* (Woods Hole Oceanographic Institution), who chased these worms for many years before finally skillfully succeeding in their capture.

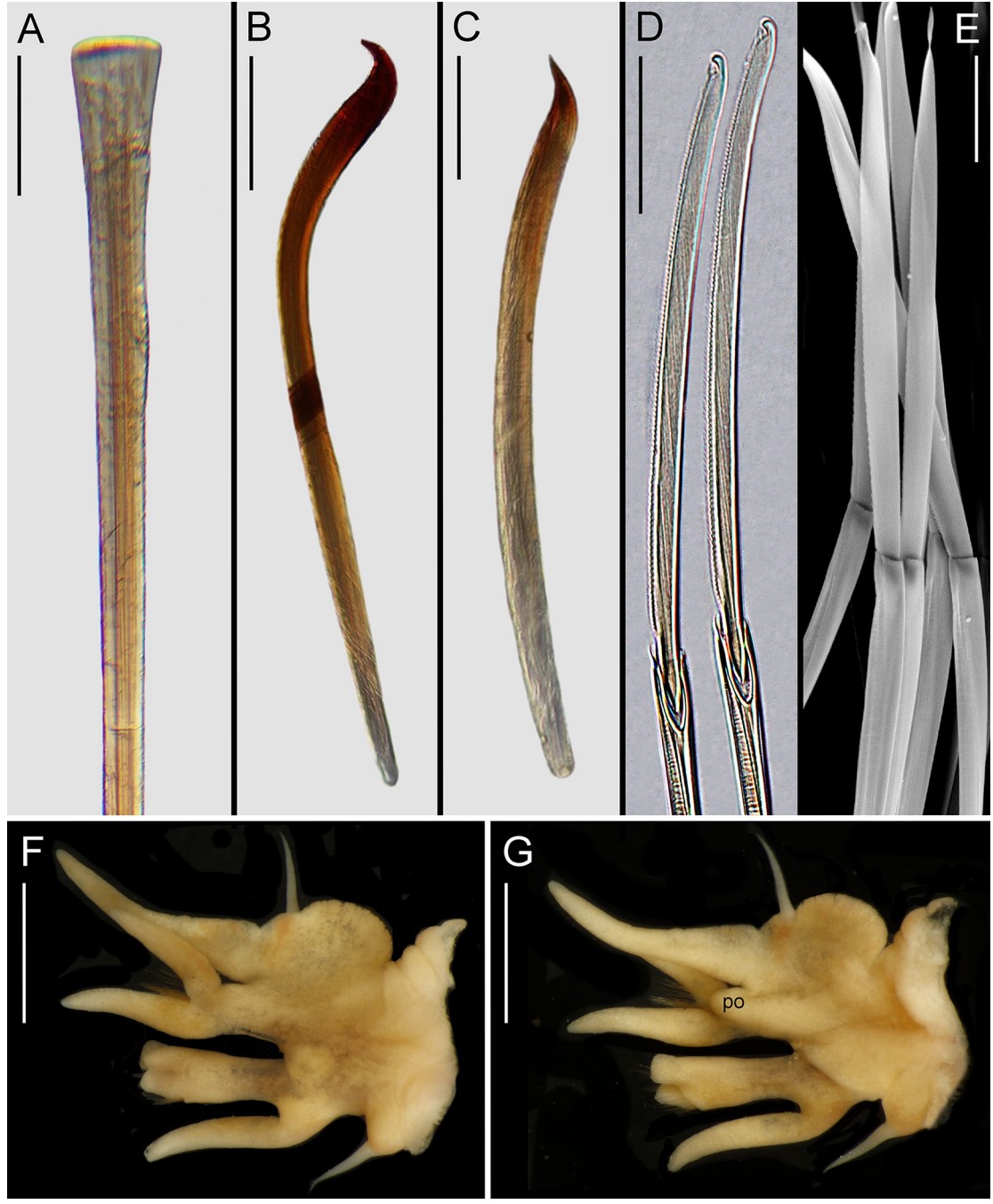

**Fig 8. *Pectinereis strickrotti* gen. nov., sp. nov. male chaetae and female parapodia.** A, D. Holotype (SIO-BIC A9836), epitokous male: A, barely expanded proximal end of notoaciculae; D, homogomph falcigers, neuropodial subacicular fascicle (chaetiger 15). B, C, E. Paratype (SIO-BIC A9889), epitokous male: B, strongly hooked notoacicula (pre-pygidial parapodia); C, hooked neuroacicula (most posterior chaetigers); E, ensiform neurochaetae from mid-natatory chaetigers. F, G. Paratype (SIO-BIC A9891), mature female: F, parapodium from possibly anterior region in anterior view; G, same parapodium in posterior view. Abbreviation: po, notopodial postchaetal lobe. Scale bars: A, B, 0.2 mm; C, 0.1 mm; D, E, 50 μm; F, G, 1 mm. Credits: A, D, Tulio F. Villalobos; B, C, E-G, Greg Rouse.

## Remarks

*Pectinereis strickrotti* gen. nov., sp. nov. is described based on epitokous males from off Costa Rica at about 1,000 m depth (Fig 1A, 1B, 1D and 1E) and a small mid-anterior fragment of an infaunal mature female (Fig 1C) sequenced for molecular analysis. As well as the DNA sequence of *COI* (Fig 2A; GenBank OQ415955), the few parapodia of this female (Figs 1C, 8F and 8G) are comparable with the other individuals of *P. strickrotti* gen. nov., sp. nov. (Fig 6E). They consisted of well-developed and slender ligules (dorsal, median, ventral, and neuroacicular) and notopodial prechaetal lobe, blunt neuropodial postchaetal, inferior and notopodial postchaetal lobes, and short, cirriform dorsal and ventral cirri (Fig 6F and 6G). Although this fragment has well-developed ova (~350 μm in diameter), it is uncertain if the female undergoes epitokal metamorphosis because the parapodia correspond seemingly to a further back anterior region, where no traces of distinct modifications are detected in nereidid females with well-developed epitoky. The holotype and three paratypes differed by a maximum of five bases (Fig 2A) across the 576 *COI* bases sequenced for all specimens, an uncorrected distance of less than 1%. An epitoke was also seen (Fig 1E and S2 Video) near the type locality in January 2019 at the Parrita Scar methane seep (8.951˚ N; 84.634˚ W) at ~1,100 m on dive S0218 of the ROV *SuBastian*. The animal was collected via slurp but escaped before reaching the surface.

## Type locality

Close to summit of Mound 12, near methane seeps at ~1,000 m depth (8.929˚ N; 84.313˚ W), off the Pacific coast of Costa Rica.

## Biology

The swimming worms collected in this study were epitokous males. One of them spawned out all the sperm after removing tissue of most-posterior end for molecular purposes. One incomplete atokous female with well-developed ova (~350 μm diameter) was found dwelling in the sediment (Fig 1C). We can infer that only the males become epitokes and the females remain in the sediment, as occurs in species of *Alitta* such as *A. grandis* [6, 99, 130] and possibly *Websterinereis glauca* (Claparède 1870) [112].

## Habitat

Near methane seeps; ranging ~1,000–1,100 m depth.

## Distribution

Species known only from type locality.

## Discussion

In the present study, we establish a new nereidid genus and species, *Pectinereis strickrotti* gen. nov., sp. nov., from the deep sea off the Pacific coast of Costa Rica using both atokous and epitokous morphological features and molecular evidence. Additionally, we determined it as a member of the subfamily Nereidinae while assessing its evolutionary position within the family using both mitogenome-scale and multi-gene phylogenetic analysis. The latter result shows no obvious close relatives based on the currently available data.

## Embracing epitokal morphology

Several nereidid genera were proposed in earlier literature based on the epitokal morphology [131–136], but they were later synonymized as represented reproductive forms of members of other genera [137, 138]. The use of epitokal morphology within nereidid systematics was largely ignored for a long period. Nonetheless, it has been recently demonstrated that the use of morphological and developmental information from epitokes not only provides valuable information to distinguish at the species level but even enables distinction at broader levels [4, 101, 136, 139–142].

For instance, *Kainonereis* Chamberlin, 1919 and *Sinonereis* Wu and Sun, 1979 were re-evaluated and distinguished from currently valid genera based on the evident epitokal modifications; for instance, the presence of the elytriform and the napiform dorsal cirri, respectively [140, 141]. Furthermore, the unique combination of non-reproductive and epitokal patterns of a species previously recognized in *Composetia* aided in distinguishing the new genus *Parasetia* Villalobos-Guerrero, Conde-Vela & Sato, 2022 from morphologically similar genera [4]. Additionally, *Neanthes* was strengthened as non-monophyletic considering the broad atoke and epitokal unevenness among its species [101]. The relationships within *Neanthes* had already pointed as polyphyletic in earlier morphological analysis [143], and later confirmed using molecular phylogenetics [15, 57].

The presence of two atokal (dorsal cirrostyles as pectinate gills and elongate dorsal ligule emerging sub-medially to enlarged cirrophores) and two epitokal (hooked aciculae and elfin-shoe shaped ventral cirrophores) are distinctive features of the remarkable deep-sea nereidid *Pectinereis strickrotti* gen. nov., sp. nov. Together with the distinct phylogenetic placement, the establishment of a new genus and the description of a new species appears well justified.

## Phylogeny and systematics

The phylogenetic reconstructions shown here were inferred from a three-gene analysis (Fig 3) and the complete newly obtained mitogenomes of three species of nereidids plus other sequences retrieved from GenBank (Fig 2B). The latter analysis showed a general agreement with the previous mitogenomic assessment of Nereididae [67], supporting the current non-monophyletic status of Nereidinae de Blainville, 1818. However, this subfamily is presently the most difficult to diagnose since its membership is broader and much more heterogeneous and speciose than the others. Despite this complexity, *Pectinereis strickrotti* gen. nov., sp. nov. is regarded within Nereidinae as shown in the topologies of molecular phylogenies and by the overall morphological characters: prostomium with anteriorly entire, biramous parapodia, single ventral cirri, and paragnaths on the pharynx.

### *Paraleonnates* and *Laeonereis*: Nereidinae or Gymnonereidinae?

The type species of *Paraleonnates* (*P. uschakovi* Chlebovisch & Wu, 1962), a supposed member of Nereidinae [see 6], showed a variable position (Figs 2B and 3), as did *Laeonereis*, which was nested inside Nereidinae (Figs 2B and 3), when it is typically referred to as a member of Gymnonereidinae [see 128]. Despite both genera being the subject of recent comprehensive morphological studies [e.g., 106, 144], none have dealt with their current subfamily position.

Gymnonereidinae was originally erected by Banse [145] for *Gymnonereis* Horst, 1919, *Ceratocephale* Malmgren, 1867, *Micronereides* Day, 1963 and *Tambalagamia* Pillai, 1961, due to the presence of two unique features among nereidids: (1) the anterior region carrying dense tufts of chaetae, and (2) the double neuropodial cirri. However, Fitzhugh [128] expanded it to include all genera without paragnaths, except Namanereidinae, encompassing thus a more complex group of taxa. Hylleberg & Nateewathana [146] and Khlebovich [6] followed Banse's

original concept. Later, Santos et al. [147] restricted Gymnonereidinae *sensu* Banse [145] to the four original genera through a phylogenetic analysis based upon morphology mainly by the presence of double ventral cirri, dense chaetal bundles in anterior parapodia, and subacicular notopodial chaetae. On the contrary, both *Paraleonnates* and *Laeonereis* have single ventral cirri, and the chaetal bundles of the anterior parapodia do not differ significantly in density from the subsequent ones. This different morphology and the molecular tree based on mitochondrial genomes support that none of these two genera belongs to Gymnonereidinae, in which *Laeonereis* was earlier proposed [see 128] but is currently positioned as sister group to Nereidinae [129, this study].

*Paraleonnates* consists presently of four species: *Paraleonnates bolus* (Hutchings & Reid, 1991), *P. sootai* (Misra, 1999), *P. tenuipalpa* (Pflugfelder, 1933), and the type species *P. uschakovi* Chlebovitsch & Wu, 1962. The genus has been typically considered within Nereidinae. Recently, morphology and molecular analysis suggested *Paraleonnates* does not belong in Nereidinae based on phylogenetic results and in having a Group II mitochondrial gene order, which differs from Group I by the position of three t-RNA genes encoding for tyrosine, methionine, and aspartic acid [61, 67, 129, 147]. In the present study, *P. uschakovi* was placed as the sister group to all other Nereidinae sampled here according to the three-gene tree (Fig 3) or sister to the clade of *Tylorrhynchus*+ Namanereidinae + Dendronereidinae based on the mitogenomic tree (Fig 2B). At present, we concur with previous studies in not placing *Paraleonnates* within Nereidinae based on these conflicting results. A critical morphological reassessment of *Paraleonnates* is needed as well as further molecular data from more taxa.

*Paraleonnates* species exhibit a unique feature among nereidids, namely that the maxillary ring of the proboscis has dorsally transverse rows of paragnaths and papillae. Contrary to the typical features present in the subfamily Nereidinae, *Paraleonnates* has a set of shared characters that makes its species distinctive: 1) prostomium anteriorly with a deep cleft, 2) glandular oesophageal caeca absent, 3) first two chaetigers with notoacicula, 4) three parapodial lobes present only in neuropodia (superior, inferior, and postchaetal), and 5) falcigers absent throughout. The first three features link *Paraleonnates* with *Ceratonereis* Kinberg, 1865 and *Solomononereis* Gibbs, 1971, two genera related to each other and presently treated as belonging to Nereidinae, although their phylogenetic position within the family is still uncertain—being part of Nereidinae according to morphology [143, 147], or doubtfully *Ceratonereis* as sister to all nereidids based on molecular markers [129]. However, *Ceratonereis* and *Solomononereis* lack a superior neuropodial lobe, present falcigers, and have different arrangement of paragnaths dorsally on the maxillary ring. On the other hand, the last three features can partially be shared with *Alitta*, *Nectoneanthes*, and a few *Neanthes* species with three neuropodial lobes. *Paraleonnates* shares with *Nectoneanthes* both the presence of notoacicula in the first two chaetigers—present in *Alitta* and absent in those *Neanthes* species—and the absence of falcigers throughout the body —absent in *Alitta* and *Neanthes* species. Nevertheless, *Paraleonnates* can be easily distinguished from the members of those three genera by the absence of a notopodial postchaetal lobe, the prostomium with an anterior cleft, and the absence of glandular caeca.

## Accepting Dendronereidinae

In this study, we also propose to reinstate Dendronereidinae Pillai, 1961 as valid. The subfamily was originally proposed to include the gill-bearing genera *Dendronereis* Peters, 1854, *Dendronereides* Southern, 1921, and *Tambalagamia* Pillai, 1961 with the former as type [98]. However, as suggested previously [1, 91, 145, 147] and discussed above, the complex gills in members of those genera are not homologous. They are markedly modified, branched, bipinnate dorsal cirrophores in *Dendronereis*, arborescent tufts inserted basally between the dorsal

cirrostyle and the median ligule in *Dendronereides*, and foliaceous dorsal cirrophore in *Tambalagamia*. The latter belongs to Gymnonereidinae due to its close morphological [145, 147] and molecular [67, 129] similarity to *Gymnonereis* and *Ceratocephale*. Also, *Dendronereis* is distinguishable from *Dendronereides* by having a ventral ligule (absent in *Dendronereides*) and in lacking notopodial glandular organs and compound falcigers (both present in *Dendronereides*). However, molecular studies of *Dendronereides* have yet to be undertaken.

Members of *Dendronereis* are unique among nereidids by the presence and type of gills restricted to some anterior chaetigers. Recently, a complete mitogenome was published for *Dendronereis chipolini* Hsueh, 2019, whose identity was confirmed by a nereidid specialist [66]. In the present results (Fig 2 and 3), the placement of *D. chipolini* was as a sister-group to an apparently misidentified specimen of the species published as *Neanthes glandicincta* [64], and, based on the sequence similarities, is likely also *D. chipolini*. The placement of *D. chipolini* supports the recognition of Dendronereidinae in contrast to its synonymy with Gymnonereidinae as previously proposed [128].

Although *Dendronereides* was not included in the present mitogenomic analysis, we suggest exclusion from Dendronereidinae because of its marked differences with *Dendronereis* and leaving it unplaced, resulting in the subfamily being monotypic as earlier proposed by Santos et al. [147]. However, its validity still needs to be addressed. *Dendronereides* is more like *Tylorrhynchus*—considered as belonging to Gymnonereidinae [128] but recently unplaced (see Figs 2B and 3; [129])—rather than other nereidid genera mainly because of the presence of papillae on both rings of the proboscis, dorsal cirrophores consisting of glandular organs with openings, compound falcigers, and the absence of ventral ligule. These genera require further morphological and molecular data to assess their placement.

Following the recent proposal of revisited definitions for the subfamilies Gymnonereidinae and Nereidinae [129], an emendation of Dendronereidinae is provided below considering the earlier and current knowledge on the morphology of the genus' members [e.g., 1, 91, 148–151].

### Dendronereidinae Pillai, 1961

Type genus. *Dendronereis* Peters, 1854.

**Diagnosis** (emended from Pillai [98])

Prostomium with anterior cleft. Two antennae. Palps with elongated palpophores and conical palpostyles. Tentacular belt length equal to or longer than length of chaetiger 1. Tentacular cirri with four pairs, distinct cirrophore present. Proboscis without paragnaths, papillae present on both rings; maxillary ring sometimes smooth. Paired esophageal caeca absent. Parapodia biramous, except first two chaetigers, uniramous (lacking notoacicula). Dorsal cirrophore of some anterior chaetigers divided into numerous branchial filaments. Chaetigers with gill-bearing parapodia multilobed, including notopodial postchaetal, and superior, inferior, and postchaetal lobes. Dorsal, median, and ventral ligules present. Single dorsal and ventral cirri. Notochaetae and neurochaetae with homogomph spinigers only. Mitochondrial gene order of Group II type as identified by Park et al. [61].

### Composition

One genus and five species: *Dendronereis aestuarina* Southern, 1921, *D. arborifera* Peters, 1854, *D. chipolini* Hsueh, 2019, *D. dayi* Misra, 1999, and *D. pinnaticirris* Grube, 1878.

### Supporting information

**S1 Fig.**
(JPG)

**S1 Table. Accession numbers for the phylogenetic analysis of COI, 16S rRNA and 18S rRNA shown in Fig 3.** Data was sourced from Alves et al. 2023 [129]. New sequences are indicated in **bold.** Most sequence IDs from BOLD are marked with #. * *Laeonereis* cf. *pandoensis* (Monro, 1938) is used here instead of *Laeonereis culveri* (Webster, 1879) since the specimen was collected in Brazil. One terminal from Alves et al. 2023 [129] was not included here, *Ceratonereis longiceratophora* Hartmann-Schröder, 1985 as the there was no 16S sequence, the COI sequence (AY583701) is actually a flabelligerid and the 18S sequence appears to be of a hesionid. The correct spelling for sequences lodged on GenBank as *Tylorrhynchus heterochaetus* is *Tylorrhynchus heterochetus*.
(DOCX)

**S1 Video. Epitokous living specimens of *Pectinereis strickrotti* gen.** nov., sp. nov. recorded at the type locality, swimming near the bottom at the Mound 12 methane seep at ~1,000m depth (8.930˚ N; 84.312˚ W) on Nov. 2, 2018 (dive 4987 of the DSV *Alvin*). Courtesy of Woods Hole Oceanographic Institute.
(MOV)

**S2 Video. A solitary epitoke living specimen of *Pectinereis strickrotti* gen. nov., sp. nov., swimming at the Parrita Scar methane seep (8.951˚ N; 84.634˚ W) on January 11, 2019, at ~1,100 m recorded via ROV *SuBastian* on dive S0218.** Courtesy of Schmidt Ocean Institute.
(MOV)

## Acknowledgments

We are grateful to the cruise party of AT42-03 for their help in collecting the specimens and to Marina McCowin and Charlotte Seid for initial sequencing and cataloging, respectively. The US National Science Foundation (NSF-OCE 0939557) supported this research. We also appreciate the generosity of Masaroni Sato (University of Kagoshima) in sharing fresh *Nectoneanthes* specimens for sequencing. TFVG was supported by CONACYT under the Postdoctoral Stays for Researchers in Mexico grant 2022(2) and appreciated the kind support of Omar Valencia Méndez (CICESE) by providing facilities and installations during the writing process of this contribution. Many thanks for the helpful comments by Torkild Bakken, Paulo Ricardo Alves and two anonymous reviewers on the initial submission. The Schmidt Ocean Institute kindly assisted with some of the publication costs.

## Author Contributions

**Conceptualization:** Sonja Huč, Ekin Tilic, Greg W. Rouse.

**Data curation:** Sonja Huč, Avery S. Hiley, Greg W. Rouse.

**Formal analysis:** Tulio F. Villalobos-Guerrero, Sonja Huč, Avery S. Hiley, Greg W. Rouse.

**Funding acquisition:** Greg W. Rouse.

**Investigation:** Tulio F. Villalobos-Guerrero, Ekin Tilic, Greg W. Rouse.

**Methodology:** Greg W. Rouse.

**Project administration:** Greg W. Rouse.

**Resources:** Greg W. Rouse.

**Supervision:** Greg W. Rouse.

**Validation:** Tulio F. Villalobos-Guerrero, Avery S. Hiley, Greg W. Rouse.

**Visualization:** Tulio F. Villalobos-Guerrero, Sonja Huč, Ekin Tilic, Greg W. Rouse.

**Writing – original draft:** Tulio F. Villalobos-Guerrero, Greg W. Rouse.

**Writing – review & editing:** Tulio F. Villalobos-Guerrero, Sonja Huč, Ekin Tilic, Avery S. Hiley, Greg W. Rouse.

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
