## [Decision Letter · Decision Letter 0]

31 Oct 2023

PONE-D-23-29948A remarkable new deep-sea nereidid (Annelida: Nereididae) with branchiaePLOS ONE

Dear Dr. Rouse,

Thank you for submitting your manuscript to PLOS ONE. After careful consideration, we feel that the manuscript has merit but does not fully meet PLOS ONE’s publication criteria as it currently stands. Therefore, we invite you to submit a revised version of the manuscript that addresses the points raised during the review process.A revision on the Remarks section of the new genus is required to include differences in the branchial morphology of Dendronereis/Dendronereides. It is recommended that the authors include in the Remarks section of the new genus a discussion about some of the unique characters (e.g. pectinate dorsal cirri and hooked acicular) being or not reproductive characters. Please submit your revised manuscript by Dec 15 2023 11:59PM. If you will need more time than this to complete your revisions, please reply to this message or contact the journal office at plosone@plos.org. Please include the following items when submitting your revised manuscript:A rebuttal letter that responds to each point raised by the academic editor and reviewer(s). You should upload this letter as a separate file labeled 'Response to Reviewers'.A marked-up copy of your manuscript that highlights changes made to the original version. You should upload this as a separate file labeled 'Revised Manuscript with Track Changes'.An unmarked version of your revised paper without tracked changes. You should upload this as a separate file labeled 'Manuscript'.If applicable, we recommend that you deposit your laboratory protocols in protocols.io to enhance the reproducibility of your results. Protocols.io assigns your protocol its own identifier (DOI) so that it can be cited independently in the future. For instructions see: https://journals.plos.org/plosone/s/submission-guidelines#loc-laboratory-protocols. Additionally, PLOS ONE offers an option for publishing peer-reviewed Lab Protocol articles, which describe protocols hosted on protocols.io. Read more information on sharing protocols at https://plos.org/protocols?utm_medium=editorial-email&utm_source=authorletters&utm_campaign=protocols.

We look forward to receiving your revised manuscript.

Kind regards,

Wagner Magalhães

Academic Editor

PLOS ONE

Journal Requirements:

We are grateful to the cruise party of AT42-03 for their help in collecting the specimens and to Marina McCowin and Charlotte Seid for initial sequencing and cataloging, respectively. The US National Science Foundation (NSF-OCE 0939557) supported this research. We also appreciate the generosity of Masaroni Sato (University of Kagoshima) in sharing fresh Nectoneanthes specimens for sequencing. TFVG was supported by CONACYT under the Postdoctoral Stays for Researchers in Mexico grant and appreciated the kind support of Omar Valencia Méndez (CICESE) by providing facilities and installations during the writing process of this contribution.

GWR NSF-OCE 0939557 US National Science Foundation https://www.nsf.gov. The funders had no role in study design, data collection and analysis, decision to publish, or preparation of the manuscript.

Additional Editor Comments:

Dear authors,

The manuscript has been reviewed by four nereidid specialists and they all agreed that it only needs minor revisions. Although all reviewers did not agued that the newly described taxon is unique and different from all other nereidid species, there is a need to expand the discussion about the proposal of new genera based on epitoke specimens, which is clearly a limitation of this study. I also suggest that the authors include this discussion in the remarks of the new genus. All four reviewers have suggested several minor edits and improvements to be addressed by the authors.

Reviewers' comments:

Reviewer's Responses to Questions

**Comments to the Author**

1. Is the manuscript technically sound, and do the data support the conclusions?

Reviewer #1: Yes

Reviewer #2: Yes

Reviewer #3: Yes

Reviewer #4: Partly

2. Has the statistical analysis been performed appropriately and rigorously? 

Reviewer #1: Yes

Reviewer #2: Yes

Reviewer #3: Yes

Reviewer #4: N/A

3. Have the authors made all data underlying the findings in their manuscript fully available?

Reviewer #1: Yes

Reviewer #2: Yes

Reviewer #3: Yes

Reviewer #4: Yes

4. Is the manuscript presented in an intelligible fashion and written in standard English?

Reviewer #1: Yes

Reviewer #2: Yes

Reviewer #3: Yes

Reviewer #4: Yes

5. Review Comments to the Author

Reviewer #1: The authors describe a new species and genus in the family Nereididae. The ms is well written, and reads well. Description of the new species is detailed, with nice and detailed illustrations and it is well justified. I have provided detailed and technical comments in the ms pdf.

The authors draw attention to the rare occasion a new nereidid genus has been described based on a few reproductive specimens, with unusual characters. This raises some questions. How unique are the characters to advocate for a new genus? This is well argued for. But, some of the characters, typically pectinate dorsal cirri/cirrostyle and hooked aciculae, are unusual among nereidids. Are they (simply?) reproductive characters in epitokes? And, subject to sexual dimorphism? This is discussed, in detail (p 27-29) in a sound discussion, but I would challenge the authors to be open more open in their discussion on this, on the morphology being different when atokous specimens will be possible to describe.

Molecular data complementing the morphological analyses is an important contribution to build a bigger picture of phylogenetic relationships in Nereididae. All that we do not understand in this group, then adding bits and pieces as they become available have great value.

Sincerely,

Torkild Bakken

Reviewer #2: The morphological descriptions are clear and terminology is adequately referenced and justified. The figures are very clear. The genus and species described is indeed new and remarkable

I am unable to comment on the molecular methods but have no doubts of their scientific rigor. The mitochondrial genome dataset strongly supports novelty of the net genus and a distant relationship with other taxa for which mito data are currently available. Validation of Dendronereidinae is also supported. It is a pity mito data are currently biased towards Nereidinae with only one nominal Gymnonereidinae genus, Tylorrhynchus (figure 2B). That biased taxon sampling is repaired by the 3 gene tree, but given the discussion in the manuscript of polyphyletic Gymnonereidinae it is a pity that result is hidden in the tree in the supplementary data. I suppose that was a space constraint from the journal?

I have only a very few minor improvements that should be considered by the authors:

on page 11, lines 210-211

"Descriptions of the species are yielded on the holotype morphology unless otherwise stated."

should simply be:

"Descriptions of the species are based on the holotype morphology unless otherwise stated."

However, in lines 219-220:

"The placement and irrigation of the notopodial vessels running alongside the lateral margins of such an enlarged structure are homologs."

Yes, very likely they are homologs but even so it would be preeferable to state that this is an assumption. Also, it is the notopodium that is being irrigated, not the notopodial vessels, which are the structures doing the irrigating. Here, "irrigating" really means form or structure.

so that sentence should be:

"The placement and structure of the notopodial vessels running alongside the lateral margins of such an enlarged structure are here assumed to be homologs."

Line 232 on the next page will need similar editing.

On page 16 line 337 there is an autocorrect typo: at the end of the line "readably" should be "readily"

Reviewer #3: The study makes an important contribution to the knowledge of the biology and systematics of Nereididae. The methodology is appropriate for the questions raised in the study and the results support the conclusions presented. The new species described presents unique characteristics that reinforce the validity of the new taxon.

The most sensitive issue in the study is the determination of a new genus and a new species based on characters related to epitoky. In my opinion, this is not necessarily wrong, but it could lead to synonymy of the genus in the future if non-epitokous individuals are found and these are similar to an already described genus. On the other hand, I see no doubt in the validity of organisms described as belonging to a new species. The systematics of Nereididae is still very confusing and the boundaries of the genera are not yet well established. I recommend including information regarding the genetic distance of the analyzed taxa, at least as supporting material, to help readers draw their conclusions. Despite all this, out of respect for the authors' opinion, I believe that the manuscript should include the description of the new genus in its final version as it is now. The Nereididae specialists community will decide whether to confirm this taxon in future studies.

Another result presented that may raise discussions is the reestablishment of Dendronereidinae based on two sequences that are, in fact, from just one species. Although it may be a bit hasty, I believe that the systematics of Nereididae needs these new definitions to overcome a classification that is widely used but does not reflect the phylogeny of the group.

Both regarding the description of the new genus and the reestablishment of Dendronereidinae, the authors make their opinions on the subject clear in the text, therefore I believe that the conclusions are appropriately defended and valid.

I recommend that the manuscript should be accepted for publication with minor revisions that I present as suggestions for enriching the study.

• The need for the haplotype network is unclear. The genetic distance between the specimens is interesting, but translating it into a haplotype network seems unnecessary to me. Either the authors should discuss this result further, presenting a hypothesis that can be related to the network, or they can remove it from the main body of the manuscript without prejudice to the main conclusions presented.

• Figure S1 is not clearly rooted. It seems to me just a matter of detail in the drawing of the figure since the analysis includes outgroups. Furthermore, I would include this figure in the main manuscript and not as supporting material, if there are no problems with limitations of figures or pages in the journal.

• Still on the figures, I would not mark Tylorrhynchus as Gymnonereidinae. following the recent work of Alves et al. (2023) this species would not be included in this subfamily. An alternative would be to make this issue clearer in the figures and in the manuscript.

• The authors dedicate a paragraph of the discussion to explaining their position on describing a taxon based on epitokous, which is great. But I suggest that this paragraph be included in the remarks of the diagnosis of the new genus. I believe this is an important aspect of the taxonomic act and including this in the remarks makes the authors' position clear from the beginning.

Reviewer #4: There is no doubt that the study presented contributes significantly to the knowledge of the diversity of the Nereididae group and the diversity of the deep sea. In addition to contributing to the phylogeny hypotheses of the Nereididae subfamilies.

It is well written, with detailed morphological description and excellent images. However, although the importance and level of information on epitokous morphology has finally been recognized in species delimitation in recent decades, the description of a new genus based only on epitokous males has its limitations. Mainly when examining the anterior parapodia of a female fragment, in the reproductive stage even if it is not known which stage, and which do not present one of the diagnostic characteristics of the genus. In addition to limiting the discussion regarding the phylogenetic position of the group based on morphology, even if the phylogenetic analysis was based on molecular data. Something that becomes clear in the discussion on phylogeny, which is more focused on the Paraleonnates taxon than Pectinereis. Of course, it does not invalidate the description of the new genus, but reinforces that it is not ideal to describe new groups only based on epitokous specimens.

Below, some little suggestions and doubts..

..

Line 33. Should be more clear that the new genus is not part of Dendronereidinae and also mention the taxa included in the new proposal, considering it differs from the original Pillai Dendronereidinae.

Line 86. Add “using epitokous specimens

Considering your proposal of ressurecting Dendronereidinae subfamily and presente a emended diagnosis I think the topic should be mentioned in the introduction.

Table 1. Laeonereis cf. pandoensis (Munro, 1938) or (Monro, 1937) ?

Platynereis cf. australis author and year?

Line 303. Ajust the font size of anterior achaetous segmental region

Line 300. The genus diagnosis is almost a description, should be shorter. Add it is based on epitokous

Line 320 -321. Hooked aciculae, characteristic mentioned in the abstract as diagnostic to the genus doens´t apear in the diagnosis. shouldn´t appear here?

Line 322. Notochaetae with? homogomph spinigers

Line 328 ..comb-like parapodial cirrostyles or pectinate .. just to follow the nomenclature adopted

Line 338. The hooked aciculae should be mentioned considering is cited as diagnostic in the abstract?

Line 341. And papillae only on both pharyngeal rings?

Line 343. You are assuming that all taxa in nereidid have a ventral cirrophore? Be more clear.

Line 365. The name of authors must be in italic?

Line 432>. Why the pharynx was only observed in the holotype? Because was everted only in this specimen you decided not dissected the paratypes?

Line 466. Branchiae?

Line 452. Describe the pre-pigidial region and Indicated in the Figure 3E, even being easy to visualize.

Line 523. Why to choose the specimen with post-natatory region lost as holotype considering you have more specimens?

Line 559. Homogomph falcigers in neuropodia are not so common, are not present in all Nereididae genera and even probably being a homoplastic feature could be pointed in the remarks or discussion.

Line 564. Blades of Homogomph falcigers spiniger-like, long. Why not only describe as long blade (ratio...)?

Line 565. Lacking incurved terminal tooth. Sounds like it was lost.

Line 577. The female parapodia in figure 7G-F doens´t present pectinate cirrostyle dorsal or ventral cirrophore. Also couldn´t see them in the figure 1C. So, the unsual diagnostic feature, the pectinate cirrostyle/cirropohore dorsal and ventral, could be present only in male epitokous?

Line 584. Sure no modifications are detected in anterior region of epitokous in any species of nereidid? Or do you mean in females?

Line 573. Remarks the text that was expected in the remarks with details about morphology is in the discussion..

Line 608. Not clear how you came to conclusion this is the first endemic deep sea nereidid.

Line 609-612- not necessary, sounds like repeating part of the first three lines of the discussion and the main goals of the paper.

Line 621.No doubt it would be better to have epitokous and atokous in all taxa descriptions,considering some previous studies already confirmed it s possible delimitate species based on epitokous morphology, but It is not always possible to get epitokous male and females specimens when describing a new taxon. I believe this is the main reason for not using their morphology for stablishing taxa nowadays.

In the case of this study, considering you have molecular data, won´t be difficult to confirm the identification of atokous specimens found in the future, of course. This is essential to complete the description of the new genus. Specially considering the difference when you compare the parapodia of epitokous male and atokous female, that doens´t show one of the the main diagnostic features of the new genus. Maybe the atokous male are morphologically diferent.

Line 670. Powerfull supportive chaetae. no need of using the adjective powerful.

Line 717. In the epitoke?

Line 719. ..generally smaller, shorter? Sounds more precise than less developed

Line 720. Pre-natatory parapodia? The epitokous term is used only for the modified parapodia or for Whole specimen modified for reproduction? Even though you have the anterior chaetigers not modified you can have modifications in the eyes, for example. So, sounds weird use the word pre-epitokous if you are describing epitokous specimens.

Line 727. All this comparisons about the dorsal ligule are related to the pre-natatory region of Pectinereis considering you are comparing to atokous specimens in the other genera or no? Make it clear. Is it indicated to compare in epitokous with atokous?

Line 748. Phylogeny and systematics: In this section I expected to see more discussion about the position of the new genus than for Paralleonnates, for example.

S1Table. “as the there was no 16S sequence, the COI sequence (AY583701) is actually a flabelligerid and the 18S sequence appears to be of a hesionid”. Apart from the mistake committed by the authors mentioned, suggest also cite the authors that included the sequences in GenBank.

Figure 1A. doens´t bring much information and B and C show what A was supposed to.

Figure 2. Tylorhynchus as Gymonereidinae ?

Figure 5. A. chaetiger 1 or 2? In the legend is on 1 and in the plate 2. I. pre-pygidial parapodia? Why not keep the same term that is in the description instead of most posterior?

Figure 7. use the term pre-pygidial instead of most posterior to keep the pattern

6. PLOS authors have the option to publish the peer review history of their article (what does this mean?). If published, this will include your full peer review and any attached files.

Reviewer #1: No

Reviewer #2: No

Reviewer #3: **Yes: **Paulo Ricardo Alves

Reviewer #4: No

---

## [Author Response · Author response to Decision Letter 0]

15 Dec 2023

Reviewer #1: The authors describe a new species and genus in the family Nereididae. The ms is well written, and reads well. Description of the new species is detailed, with nice and detailed illustrations and it is well justified. I have provided detailed and technical comments in the ms pdf.

The authors draw attention to the rare occasion a new nereidid genus has been described based on a few reproductive specimens, with unusual characters. This raises some questions. How unique are the characters to advocate for a new genus? This is well argued for. But, some of the characters, typically pectinate dorsal cirri/cirrostyle and hooked aciculae, are unusual among nereidids. Are they (simply?) reproductive characters in epitokes? And, subject to sexual dimorphism? This is discussed, in detail (p 27-29) in a sound discussion, but I would challenge the authors to be open more open in their discussion on this, on the morphology being different when atokous specimens will be possible to describe.

R: Thank you. Additional comments were provided in the results and discussion of the manuscript to address in further detail these interrogatives.

Molecular data complementing the morphological analyses is an important contribution to build a bigger picture of phylogenetic relationships in Nereididae. All that we do not understand in this group, then adding bits and pieces as they become available have great value.

Sincerely,

Torkild Bakken

Reviewer #2: The morphological descriptions are clear and terminology is adequately referenced and justified. The figures are very clear. The genus and species described is indeed new and remarkable

I am unable to comment on the molecular methods but have no doubts of their scientific rigor. The mitochondrial genome dataset strongly supports novelty of the net genus and a distant relationship with other taxa for which mito data are currently available. Validation of Dendronereidinae is also supported. It is a pity mito data are currently biased towards Nereidinae with only one nominal Gymnonereidinae genus, Tylorrhynchus (figure 2B). That biased taxon sampling is repaired by the 3 gene tree, but given the discussion in the manuscript of polyphyletic Gymnonereidinae it is a pity that result is hidden in the tree in the supplementary data. I suppose that was a space constraint from the journal?

I have only a very few minor improvements that should be considered by the authors:

on page 11, lines 210-211

"Descriptions of the species are yielded on the holotype morphology unless otherwise stated."

should simply be:

"Descriptions of the species are based on the holotype morphology unless otherwise stated."

Changed

However, in lines 219-220:

"The placement and irrigation of the notopodial vessels running alongside the lateral margins of such an enlarged structure are homologs."

Yes, very likely they are homologs but even so it would be preeferable to state that this is an assumption. Also, it is the notopodium that is being irrigated, not the notopodial vessels, which are the structures doing the irrigating. Here, "irrigating" really means form or structure.

so that sentence should be:

"The placement and structure of the notopodial vessels running alongside the lateral margins of such an enlarged structure are here assumed to be homologs."

Line 232 on the next page will need similar editing.

R: Thanks. All comments mentioned above were followed

On page 16 line 337 there is an autocorrect typo: at the end of the line "readably" should be "readily"

Changed

Reviewer #3: The study makes an important contribution to the knowledge of the biology and systematics of Nereididae. The methodology is appropriate for the questions raised in the study and the results support the conclusions presented. The new species described presents unique characteristics that reinforce the validity of the new taxon.

The most sensitive issue in the study is the determination of a new genus and a new species based on characters related to epitoky. In my opinion, this is not necessarily wrong, but it could lead to synonymy of the genus in the future if non-epitokous individuals are found and these are similar to an already described genus. On the other hand, I see no doubt in the validity of organisms described as belonging to a new species. The systematics of Nereididae is still very confusing and the boundaries of the genera are not yet well established. I recommend including information regarding the genetic distance of the analyzed taxa, at least as supporting material, to help readers draw their conclusions. Despite all this, out of respect for the authors' opinion, I believe that the manuscript should include the description of the new genus in its final version as it is now. The Nereididae specialists community will decide whether to confirm this taxon in future studies.

Another result presented that may raise discussions is the reestablishment of Dendronereidinae based on two sequences that are, in fact, from just one species. Although it may be a bit hasty, I believe that the systematics of Nereididae needs these new definitions to overcome a classification that is widely used but does not reflect the phylogeny of the group.

Both regarding the description of the new genus and the reestablishment of Dendronereidinae, the authors make their opinions on the subject clear in the text, therefore I believe that the conclusions are appropriately defended and valid.

Thankyou

I recommend that the manuscript should be accepted for publication with minor revisions that I present as suggestions for enriching the study.

• The need for the haplotype network is unclear. The genetic distance between the specimens is interesting, but translating it into a haplotype network seems unnecessary to me. Either the authors should discuss this result further, presenting a hypothesis that can be related to the network, or they can remove it from the main body of the manuscript without prejudice to the main conclusions presented.

We have left the COI haplotype network in place as there is a clear space in the figure and it gives a quick visual confirmation that the female atokous specimen and the three male epitokes that were sampled are the same species with little variation on COI

• Figure S1 is not clearly rooted. It seems to me just a matter of detail in the drawing of the figure since the analysis includes outgroups. Furthermore, I would include this figure in the main manuscript and not as supporting material, if there are no problems with limitations of figures or pages in the journal.

We moved the figure into the main manuscript as Figure 3 and renumbered the other Figures accordingly. We did state that in the M&M and Table S1 that Chrysopetalidae terminals were used to root the tree but left them out for ease of viewing. We have now included the outgroups. 

• Still on the figures, I would not mark Tylorrhynchus as Gymnonereidinae. following the recent work of Alves et al. (2023) this species would not be included in this subfamily. An alternative would be to make this issue clearer in the figures and in the manuscript.

Tylorrhynchus is longer marked as Gymnonereidinae in either of the phylogenetic figures (2, 3).

• The authors dedicate a paragraph of the discussion to explaining their position on describing a taxon based on epitokous, which is great. But I suggest that this paragraph be included in the remarks of the diagnosis of the new genus. I believe this is an important aspect of the taxonomic act and including this in the remarks makes the authors' position clear from the beginning.

R: Thanks for your recommendation. Remarks in the genus taxonomic section should be treated for comparison purposes between other closely related genera, and that’s why we included the paragraphs in ‘Discussion’. However, we discussed the ‘Haplotype network and phylogenetic position’ and the ‘Morphological features’ in separate sections of ‘Results’. Most of the discussion on morphology was moved in order to have a detailed background on the morphological features that aided in establishing Pectinereis.

Reviewer #4: There is no doubt that the study presented contributes significantly to the knowledge of the diversity of the Nereididae group and the diversity of the deep sea. In addition to contributing to the phylogeny hypotheses of the Nereididae subfamilies.

It is well written, with detailed morphological description and excellent images. However, although the importance and level of information on epitokous morphology has finally been recognized in species delimitation in recent decades, the description of a new genus based only on epitokous males has its limitations. Mainly when examining the anterior parapodia of a female fragment, in the reproductive stage even if it is not known which stage, and which do not present one of the diagnostic characteristics of the genus. In addition to limiting the discussion regarding the phylogenetic position of the group based on morphology, even if the phylogenetic analysis was based on molecular data. Something that becomes clear in the discussion on phylogeny, which is more focused on the Paraleonnates taxon than Pectinereis. Of course, it does not invalidate the description of the new genus, but reinforces that it is not ideal to describe new groups only based on epitokous specimens.

Below, some little suggestions and doubts..

Line 33. Should be more clear that the new genus is not part of Dendronereidinae and also mention the taxa included in the new proposal, considering it differs from the original Pillai Dendronereidinae.

R: Thank you. Done

Line 86. Add “using epitokous specimens

We did have a fragment of an atokous infaunal female

Considering your proposal of ressurecting Dendronereidinae subfamily and presente a emended diagnosis I think the topic should be mentioned in the introduction.

Now mentioned

Table 1. Laeonereis cf. pandoensis (Munro, 1938) or (Monro, 1937) ?

Platynereis cf. australis author and year?

R: Authorities of species treated as cf. were removed from the table

Line 303. Adjust the font size of anterior achaetous segmental region

Done

Line 300. The genus diagnosis is almost a description, should be shorter. Add it is based on epitokous

R: Thank you, done. Diagnosis and description of the genus are given. Epitokous features are also included, but not exclusively.

Line 320 -321. Hooked aciculae, characteristic mentioned in the abstract as diagnostic to the genus doens´t apear in the diagnosis. shouldn´t appear here?

R: Thanks, already included

Line 322. Notochaetae with? homogomph spinigers

R: Yes, notochaetae have homogomph spinigers

Line 328 ..comb-like parapodial cirrostyles or pectinate .. just to follow the nomenclature adopted

R: Thank you, edited

Line 338. The hooked aciculae should be mentioned considering is cited as diagnostic in the abstract?

R: Done

Line 341. And papillae only on both pharyngeal rings?

R: Edited

Line 343. You are assuming that all taxa in nereidid have a ventral cirrophore? Be more clear.

R: This was already mentioned in Discussion. See comments now in ‘Morphological features’ section of Results.

Line 365. The name of authors must be in italic?

Made non-italic

Line 432>. Why the pharynx was only observed in the holotype? Because was everted only in this specimen you decided not dissected the paratypes?

R: Only 3 individuals have been collected since first sighted. Preservation of the remaining two intact individuals was the main reason so one could go intact to Costa Rica and the other paratype A9889 was dissected to show the buccal region but we chose not open the pharynx givben the details of the holotype. 

Line 452. Describe the pre-pigidial region and Indicated in the Figure 3E, even being easy to visualize.

R: Pre-pygidial region has been described with further detail

Line 466. Branchiae?

Edited

Line 523. Why to choose the specimen with post-natatory region lost as holotype considering you have more specimens?

R: This was the best-preserved specimen, even when incomplete and it had the pharynx everted.

Line 559. Homogomph falcigers in neuropodia are not so common, are not present in all Nereididae genera and even probably being a homoplastic feature could be pointed in the remarks or discussion.

R: Thanks for pointing this out. Agree, some comments were also added to the remarks.

Line 564. Blades of Homogomph falcigers spiniger-like, long. Why not only describe as long blade (ratio...)?

R: Thanks, edited.

Line 565. Lacking incurved terminal tooth. Sounds like it was lost.

R: Thanks, edited.

Line 577. The female parapodia in figure 7G-F doens´t present pectinate cirrostyle dorsal or ventral cirrophore. Also couldn´t see them in the figure 1C. So, the unsual diagnostic feature, the pectinate cirrostyle/cirropohore dorsal and ventral, could be present only in male epitokous?

R: Thanks. This was discussed in species remarks. It is uncertain if the infaunal female undergoes epitokal metamorphosis because the parapodia correspond to an anterior-medial region, where no traces of distinct modifications are detected in epitokous nereidids. 

Line 584. Sure no modifications are detected in anterior region of epitokous in any species of nereidid? Or do you mean in females?

R: Thanks, rephrased.

Line 573. Remarks the text that was expected in the remarks with details about morphology is in the discussion..

R: Thanks, moved to Results.

Line 608. Not clear how you came to conclusion this is the first endemic deep sea nereidid.

R: Thanks, removed.

Line 609-612- not necessary, sounds like repeating part of the first three lines of the discussion and the main goals of the paper.

R: Thanks, edited.

Line 621. No doubt it would be better to have epitokous and atokous in all taxa descriptions, considering some previous studies already confirmed it s possible delimitate species based on epitokous morphology, but It is not always possible to get epitokous male and females specimens when describing a new taxon. I believe this is the main reason for not using their morphology for stablishing taxa nowadays. In the case of this study, considering you have molecular data, won´t be difficult to confirm the identification of atokous specimens found in the future, of course. This is essential to complete the description of the new genus. Specially considering the difference when you compare the parapodia of epitokous male and atokous female, that doens´t show one of the the main diagnostic features of the new genus. Maybe the atokous male are morphologically diferent.

R: Thanks. We respect your opinion but we do not fully follow it. Not all nereidids undergo epitokal metamorphosis, many species from different genera reproduce in the atokous stage. Thus, it is not a premise for all taxa descriptions as suggested by the reviewer. It is difficult to know whether a new species collected solely with atokous individuals undergoes indeed epitokous stages or vice versa. However, this has not been an impediment in nereidids systematics to describe new species or genera. Of course, synonyms arose from species/genera described based on epitokous individuals, but the same occurred when described as atokous. So, this prevents us from fixing an absolute position for describing taxa using either of the two stages separately. All nereidids undergoing epitoky preserve unmodified diagnostic characters. These characters are informative in distinguishing taxa and also relevant to aid in linking atokous forms with reproductive specimens of a determinate taxon. The combination of non-metamorphosed and epitokous diagnostic characters makes not only species but also genera unique, even more so when they have unique reproductive characters as in Pectinereis. Our perspective, hence, integrates the available morphological evidence present in the overall diagnostic features, including both epitokous and non-metamorphosed structures, to describe and distinguish species/genera. On the other hand, the parapodia of the female are very similar to the unmodified parapodia of males, and the molecular evidence aided in determining they belong to the same species.

Line 670. Powerfull supportive chaetae. no need of using the adjective powerful.

R: Changed

Line 717. In the epitoke?

R: Edited, thank you.

Line 719. ..generally smaller, shorter? Sounds more precise than less developed

R: Edited, thank you.

Line 720. Pre-natatory parapodia? The epitokous term is used only for the modified parapodia or for Whole specimen modified for reproduction? Even though you have the anterior chaetigers not modified you can have modifications in the eyes, for example. So, sounds weird use the word pre-epitokous if you are describing epitokous specimens.

R: The correct term is pre-natatory. We use it wrongly in a few paragraphs. Thanks for pointing this out.

Line 727. All this comparisons about the dorsal ligule are related to the pre-natatory region of Pectinereis considering you are comparing to atokous specimens in the other genera or no? Make it clear. Is it indicated to compare in epitokous with atokous?

R: Thank you, edited. The comparisons of the dorsal ligule + dorsal cirrophore are related to the posterior end as pointed out in the manuscript. 

Line 748. Phylogeny and systematics: In this section I expected to see more discussion about the position of the new genus than for Paralleonnates, for example.

The support values are very low so there is little to be said apart from it belonging within Nereidinae and there are no obvious close relatives. Apart from the mitogenome analysis we widely sampled across Nereididae for the COI, 16S,18S analysis and neither showed any close relatives. This, and the morphology, allowed us to argue for a new genus.

S1Table. “as the there was no 16S sequence, the COI sequence (AY583701) is actually a flabelligerid and the 18S sequence appears to be of a hesionid”. Apart from the mistake committed by the authors mentioned, suggest also cite the authors that included the sequences in GenBank.

It is obvious who the authors once the sequences are looked up in GenBank. We see no point in naming them in this paper.

Figure 1A. doens´t bring much information and B and C show what A was supposed to.

Fig. 1A is included because it represents the first time that the worms were sighted in 2009 but were not actually collected until much later.

Figure 2. Tylorhynchus as Gymonereidinae ?

Removed

Figure 5. A. chaetiger 1 or 2? In the legend is on 1 and in the plate 2. I. pre-pygidial parapodia? Why not keep the same term that is in the description instead of most posterior?

R: Thank you. Modified accordingly.

Figure 7. use the term pre-pygidial instead of most posterior to keep the pattern

R: Thank you, done.

Reviewer #1: Torkild Bakken

Reviewer #2: No

Reviewer #3: Paulo Ricardo Alves

Reviewer #4: No

---

## [Editor Report · Decision Letter 1]

16 Jan 2024

A remarkable new deep-sea nereidid (Annelida: Nereididae) with gills

PONE-D-23-29948R1

Dear Dr. %Rouse%,

We’re pleased to inform you that your manuscript has been judged scientifically suitable for publication and will be formally accepted for publication once it meets all outstanding technical requirements.

Kind regards,

Wagner Magalhães

Academic Editor

PLOS ONE

Additional Editor Comments (optional):

Dear authors,

Many thanks for your commitment to include the suggestions provided by the four reviewers. I have gone through the revised manuscript and have no additional changes to be made, except to add the museum registration number for the paratype deposited at the MZUCR. Please proceed with this addition at your earliest convenience.

Wagner Magalhães

---

## [Editor Report · Acceptance letter]

12 Feb 2024

PONE-D-23-29948R1 

PLOS ONE

Dear Dr. Rouse, 

I'm pleased to inform you that your manuscript has been deemed suitable for publication in PLOS ONE. Congratulations! Your manuscript is now being handed over to our production team.

Kind regards, 

on behalf of

Dr. Wagner Magalhães 

Academic Editor

PLOS ONE